# Robo recruitment of the Wave regulatory complex plays an essential and conserved role in midline repulsion

**Karina Chaudhari[1], Madhavi Gorla[1], Chao Chang[2,3], Artur Kania[2,3], Greg J Bashaw[1]***

[1]Department of Neuroscience, Perelman School of Medicine, University of Pennsylvania, Philadelphia, United States; [2]Institut de recherches cliniques de Montréal (IRCM), Montréal, Canada; [3]Department of Anatomy and Cell Biology and Division of Experimental Medicine, McGill University, Montréal, Canada

**Abstract** The Roundabout (Robo) guidance receptor family induces axon repulsion in response to its ligand Slit by inducing local cytoskeletal changes; however, the link to the cytoskeleton and the nature of these cytoskeletal changes are poorly understood. Here, we show that the heteropentameric Scar/Wave Regulatory Complex (WRC), which drives Arp2/3-induced branched actin polymerization, is a direct effector of Robo signaling. Biochemical evidence shows that Slit triggers WRC recruitment to the Robo receptor's WRC-interacting receptor sequence (WIRS) motif. In Drosophila embryos, mutants of the WRC enhance Robo1-dependent midline crossing defects. Additionally, mutating Robo1's WIRS motif significantly reduces receptor activity in rescue assays in vivo, and CRISPR-Cas9 mutagenesis shows that the WIRS motif is essential for endogenous Robo1 function. Finally, axon guidance assays in mouse dorsal spinal commissural axons and gain-of-function experiments in chick embryos demonstrate that the WIRS motif is also required for Robo1 repulsion in mammals. Together, our data support an essential conserved role for the WIRS-WRC interaction in Robo1-mediated axon repulsion.

**\*For correspondence:**
gbashaw@pennmedicine.upenn.edu

**Competing interests:** The authors declare that no competing interests exist.

## Introduction

The brain is the most complex organ in the body, with trillions of specific synapses whose formation depends on the precise targeting of axons and dendrites during nervous system development. Axons are guided to their appropriate targets by a number of conserved guidance cues and their receptors, which enable neurons to form specific connections and establish functional neural circuits. The axon guidance receptors that mediate axonal guidance and targeting are tightly regulated to achieve a controlled balance between attractive and repulsive signaling, and disruption of this process results in a number of movement disorders and other neurological deficits (*Bosley et al., 2005*; *Depienne et al., 2011*; *Jen et al., 2004*). Specifically, the Roundabout (Robo) family of repulsive axon guidance receptors has been implicated in many neurodevelopmental disorders like autism spectrum disorder, dyslexia, horizontal gaze palsy, and others (*Anitha et al., 2008*; *Hannula-Jouppi et al., 2005*; *Jen et al., 2004*; *Suda et al., 2011*). Elucidating the mechanisms by which these guidance receptors function is crucial for understanding the formation of neural circuits both during development and in disease pathogenesis.

The Drosophila midline is analogous to the vertebrate spinal cord and serves as an intermediate target for commissural axons that cross from one side of the body to the other (*Klämbt et al., 1991*; *Seeger et al., 1993*). The Drosophila ventral nerve cord has a ladder-like structure consisting of 13 repeated segments, each containing an anterior commissure and a posterior commissure into which commissural neurons extend their axons to cross the midline. Midline glial cells secrete a number of

**eLife digest** The brain is the most complex organ in the body. It contains billions of nerve cells, also known as neurons, with trillions of precise and specific connections, but how do these neurons know where to go and which connections to make as the brain grows? Neurons contain a small set of proteins known as guidance receptors. These receptors respond to external signals that can be attractive or repulsive. They instruct neurons to turn towards, or away from, the source of a signal. During embryonic development, neurons use these signals as guideposts to find their way to their destination.

One such guidance receptor-signal pair consists of a receptor called Roundabout, also known as Robo, and its cue, Slit. Robo, which is located on the neuron's surface, responds to the presence of Slit in the environment, by initiating a set of signalling events that instruct neurons to turn away. Neurons make the turn by rearranging their internal scaffolding, a network of proteins called the actin cytoskeleton. How Robo triggers this rearrangement is unclear. One possibility relies on a group of proteins called the WAVE regulatory complex, or the WRC for short. Researchers have already linked the WRC to nerve cell guidance, showing that it can trigger the growth of new filaments in the actin cytoskeleton. Proteins can activate the WRC by binding to it using a set of amino acids called a WRC-interacting receptor sequence, or WIRS for short, which Robo has.

Chaudhari et al. used fruit flies to find out how Robo and the WRC interact. The experiments revealed that when Slit binds to Robo on the outside of a nerve cell, the WRC binds to Robo via its WIRS sequence on the inside of the cell. This attracts proteins inside the cell involved in rearranging the actin cytoskeleton. Disrupting this interaction by mutating either WRC or WIRS leads to severe errors in pathfinding, because when the WRC cannot connect to Robo, neurons cannot find their way. Experiments in mouse and chicken embryos showed that vertebrates use the WIRS sequence too, indicating that evolution has conserved this method of passing signals from Robo to the cytoskeleton.

The fact that Slit and Robo work in the same way across fruit flies and vertebrates has implications for future medical research. Further work could explain how the brain and nervous system develop, and what happens when development goes wrong, but Slit and Robo control more than just nerve cell pathfinding. Research has linked disruptions in both proteins to many types of cancer, so a better understanding of how Robo interacts with the WRC could lead to new developments in different fields.

guidance cues that act on their cognate receptors present on axon growth cones to induce attraction toward or repulsion away from the midline. Slit is secreted by midline glia and acts as a repulsive ligand for the Robo family of receptors (*Brose et al., 1999*; *Kidd et al., 1999*; *Kidd et al., 1998*). There are three Robo receptors in Drosophila and four in vertebrates. The Robo receptors are transmembrane proteins with an ectodomain consisting of five immunoglobulin-like domains and three fibronectin repeats, and an intracellular domain containing short, highly conserved cytoplasmic (CC) motifs (*Bashaw et al., 2000*; *Kidd et al., 1998*). Robo1 induces repulsion in growth cones of navigating axons primarily by modulating the actin cytoskeletal network. Previous work has identified some downstream effectors for Robo1 including Ena, an uncapping protein for actin filaments (*Bashaw et al., 2000*), and Son of Sevenless (SOS), a GEF for Rac1 (*Yang and Bashaw, 2006*). However, downstream signaling of Robo1 is not completely understood, especially in relation to effectors that directly link Robo1 to the actin cytoskeleton and the nature of cytoskeletal changes orchestrated by Robo1. While it seems intuitive for repulsive signaling to induce depolymerization of the actin network, a recent study reports that dorsal root ganglion axons first extend actin-rich filopodia toward a source of Slit before retracting away from it *McConnell et al., 2016*. This challenges the prevailing notion that repulsive signaling primarily relies on actin depolymerization and suggests that the actin rearrangements occurring downstream of Robo1 are more nuanced and complex than previously thought. Indeed, several of the well-known downstream effectors of Robo1 signaling, namely Ena and Rac1, are documented enhancers of actin polymerization (*Barzik et al., 2005*; *Ridley et al., 1992*).

The Scar or WAVE regulatory complex (WRC) is a heteropentameric complex consisting of five different proteins: Scar/WAVE, CYFIP/Sra1, Kette/Nap1, HSPC300/Brick1, and Abi (*Eden et al., 2002*). Scar or WAVE contains a VCA (verprolin homology, cofilin homology, acidic) region and serves as a nucleation-promoting factor for Arp2/3, thereby driving branched actin polymerization. While mammals have multiple orthologs of these proteins, Drosophila has single homologs of all five members of the complex, making it a simpler, more tractable model system for studying the WRC. The WRC has been previously implicated in axon guidance and targeting in Drosophila and *Caenorhabditis elegans* (*Shakir et al., 2008*; *Stephan et al., 2011*; *Xu and Quinn, 2012*); however, if and how it is recruited and activated downstream of guidance receptors is not known. Recent work identified a unique binding site for the WRC known as the WRC-interacting receptor sequence (WIRS) motif (*Chen et al., 2014a*). The WIRS motif is a short six amino acid peptide sequence characterized by a bulky hydrophobic residue at position 1 and a threonine or a serine at position 3, followed by a phenylalanine at position 4. The WIRS motif is present in a number of transmembrane proteins including Robo1 (*Chen et al., 2014a*). Robo1 has a WIRS motif between its CC0 and CC1 domains that is conserved across species, including humans. Previously, the WIRS motif has been shown to be important for recruitment of the WRC by neuroligins and SYG-1 in synapse formation (*Chia et al., 2014*; *Xing et al., 2018*) and for neogenin function in maintaining the stability of adherens junctions (*Lee et al., 2016*). To our knowledge, this study is the first to demonstrate a role for the WIRS-WRC interaction in axon guidance.

Here, we show that the WRC is required for Slit-Robo1 repulsive signaling at the Drosophila midline. We present evidence that Robo1 interacts with the WRC partially via its WIRS motif and that this interaction is enhanced in the presence of Slit. We show that the WIRS motif in Robo1 is important for its ability to induce ectopic repulsion in vivo. Using rescue assays, we show that Robo1 also requires its WIRS motif to mediate repulsion in ipsilateral axons in vivo. In addition, using CRISPR-Cas9-mediated mutagenesis, we show that the WIRS motif is important for endogenous Robo1 function as mutating the endogenous WIRS motif results in the complete loss of Robo1 repulsion at the midline. Finally, we use mouse dorsal spinal cord explants and growth cone collapse assays in mouse commissural neurons, together with gain-of-function experiments in chick embryos, to demonstrate that the WIRS motif is also important for vertebrate Robo1 repulsive signaling. We propose a model in which Slit binding induces recruitment of the WRC to the WIRS motif of Robo1 where it functions in Robo1-mediated repulsion at the midline.

## Results

### The WRC interacts genetically with Slit, Robo1, and SOS

WRC members are enriched in the Drosophila ventral nerve cord during embryonic stages 12–17, encompassing the developmental window when midline crossing decisions are being made (*Schenck et al., 2004*). To confirm these previously published observations, we examined the expression of Scar by immunofluorescence and observed strong axonal staining throughout embryonic stages when midline axon guidance occurs (*Figure 1—figure supplement 1A*). To investigate the potential role of the WRC in Slit-Robo repulsion at the midline, we tested for genetic interactions between *cyfip* and *hspc300*, two members of the WRC, and the Slit-Robo signaling pathway. In wild-type embryos, FasII-positive ipsilateral axons project longitudinally and never cross the midline (*Figure 1A*). In *robo1* mutants, axons in the medial most Fas-II bundle frequently cross and re-cross the midline, resulting in a very strong ectopic crossing phenotype (*Kidd et al., 1998*; *Figure 1B*). In *slit, robo1/+* embryos, where the *slit* and *robo1* gene dosage is reduced by half, the phenotype is milder (*Figure 1D*). This represents a sensitized background in which we can detect enhancers or suppressors of the Slit-Robo pathway (*Chance and Bashaw, 2015*; *Coleman et al., 2010*; *Fan et al., 2003*; *Hsouna et al., 2003*). While we see no crossing errors in FasII-positive axons in *hspc300* mutants alone (*Figure 1C*), in the *slit, robo1/+* sensitized background, *hspc300* mutants exhibit a significant enhancement of the ectopic crossing defects (*Figure 1E*). These interactions are dosage sensitive as removing one copy of *hspc300* results in a moderate enhancement of crossing errors while removing both copies of *hspc300* results in a much stronger phenotype (*Figure 1F*). Similarly, we see almost no crossing errors in FasII-positive axons in *cyfip* mutants alone (*Figure 1G*); however, in the *slit, robo1/+* sensitized background (*Figure 1H*), *cyfip* mutants show a strong dose-dependent

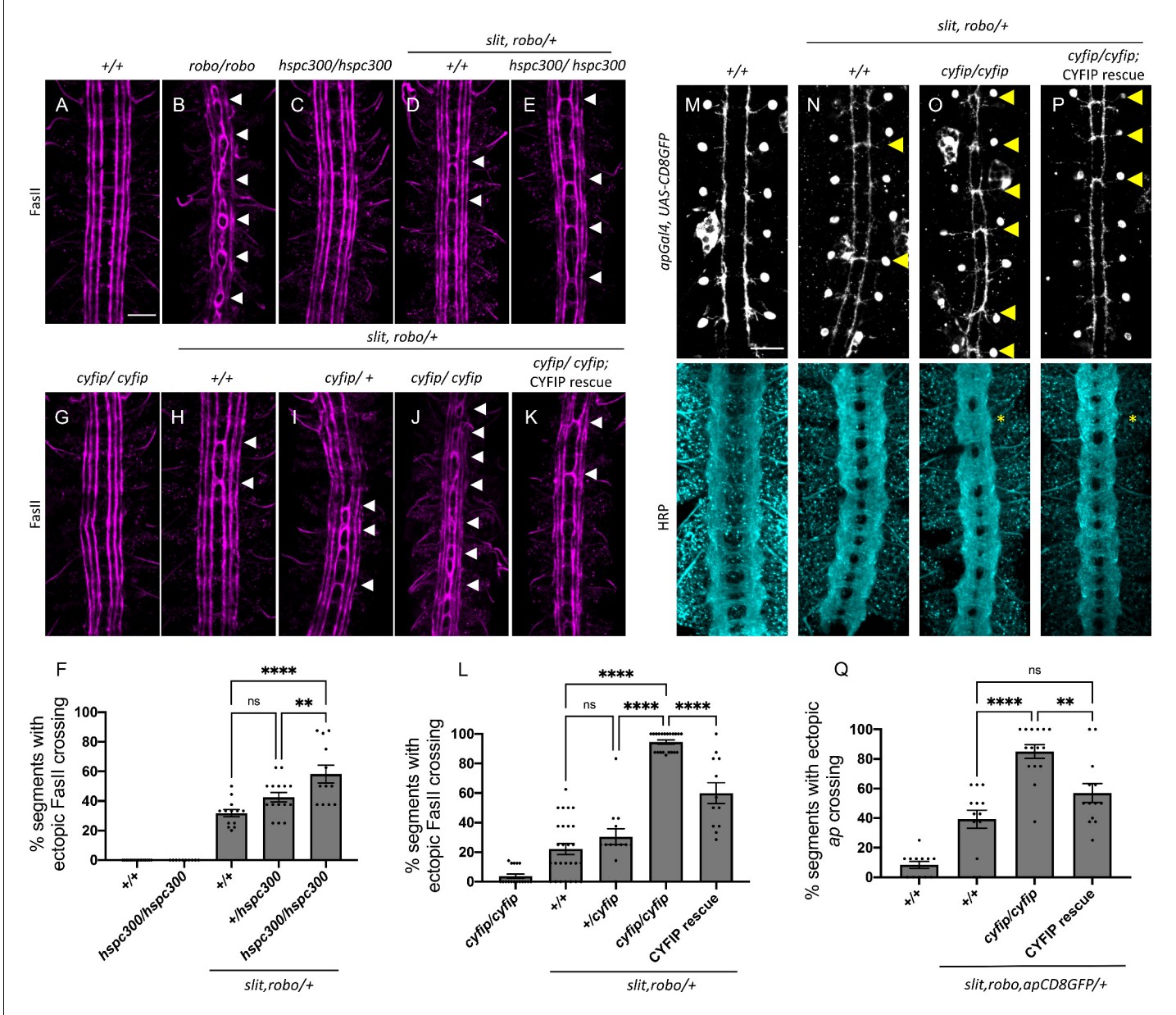

**Figure 1.** The *wave regulatory complex* genetically interacts with *slit and robo*. (A–E, G–K) Stage 17 Drosophila embryos stained with anti-FasII to label ipsilateral axons. (A, C) Wild-type and *hspc300* homozygous mutant embryos show three FasII-positive tracts that do not cross the midline. (B) *Robo* homozygous mutants show severe ectopic FasII crossing defects in 100% of segments (white arrowheads). (D) *Slit, robo* transheterozygous embryos show a mild loss-of-repulsion phenotype with ectopic FasII crossing in 31% of nerve cord segments. (E) *Hscp300* homozygous mutants that are simultaneously heterozygous for *slit* and *robo* show ectopic FasII crossing in significantly more segments of the nerve cord (58%). (G) *Cyfip* embryos have almost no ectopic crossing defects and appear like wild-type embryos. (H) Double heterozygous *slit, robo* embryos show a mild loss-of-repulsion phenotype with ectopic FasII crossing in 22% of nerve cord segments. Removing (I) one and (J) two copies of *cyfip* in a *slit, robo* background results in a dose-dependent enhancement of the ectopic FasII crossing defects (30 and 95%, respectively). (K) Driving *UAS-CYFIP* expression in neurons using the pan-neuronal *elav-Gal4* driver results in a partial rescue of the ectopic FasII crossing defects (60%). (F, L) Quantitation shows the percentage of segments in which FasII axons ectopically cross the midline. Data are presented as mean ± SEM, number of embryos, n = 15, 10, 15, 15, 12 (for F) and 17, 27, 13, 21, 12 (for L). Significance was assessed using one-way ANOVA with Tukey's multiple comparisons test. (M–P) Stage 17 embryos carrying *apGal4* and *UAS-CD8GFP* transgenes stained with anti-GFP, which labels the apterous (ap) cell bodies and axons, and anti-HRP, which labels all central nervous system (CNS) axons. (M) Wild-type embryos show ap axons that normally project ipsilaterally without crossing the midline. (N) Double heterozygous *slit, robo* embryos show a mild ectopic ap crossing phenotype of 39% (yellow arrowheads) while HRP depicts a wild type arrangement of longitudinal and commissural axon pathways. (O) *Cyfip* homozygous mutants in a *slit, robo* background show a strong enhancement of the ectopic ap crossing defects to 85% and HRP shows abnormal thickening and fusion of the commissures (asterisk). (P) Ap-specific expression of *UAS-CYFIP*

*Figure 1 continued on next page*

*Figure 1 continued*

significantly rescues the ectopic ap crossing defects (57%) but not the pan-neuronal HRP defects. (**Q**) Quantitation shows percentage of segments with ectopic apterous crossing defects. Data are presented as mean ± SEM, number of embryos, n = 12, 13, 15, 13. Significance was assessed using one-way ANOVA with Tukey's multiple comparisons test. Scale bars in (**A**) and (**M**) represent 20 µm.

The online version of this article includes the following figure supplement(s) for figure 1:

**Figure supplement 1.** Scar expression in wild-type and *scar* mutant embryos.

enhancement of the ectopic crossing defects (*Figure 1I, J*). Strikingly, removing both copies of *cyfip* in this background results in a very strong phenotype with ectopic crossing defects in nearly 100% of segments, similar to the *robo1* mutant phenotype (*Figure 1B, J*). These ectopic crossing defects can be significantly rescued by the transgenic expression of *UAS-CYFIP* using the pan-neuronal *elav-Gal4* driver (*Figure 1K, L*). This suggests that the neuronal function of CYFIP is important for Slit-Robo-mediated repulsion at the midline. It is important to note that zygotic *hspc300* and *cyfip* mutants, like mutants for all other members of the WRC, still have significant amounts of the protein remaining due to maternal deposition (*Schenck et al., 2004*; *Zallen et al., 2002*). This likely explains why these zygotic mutants have no phenotype on their own. This can be seen in *scar* zygotic mutants where the overall Scar protein level is significantly reduced but there is still a considerable amount of Scar protein remaining in central nervous system (CNS) axons (*Figure 1—figure supplement 1B, C*).

To determine whether CYFIP is required cell-autonomously, we examined a more restricted subset of ipsilateral axons, the apterous (ap) axons. Just like FasII axons, ap axons are sensitive to a partial loss of repulsion. Reducing the *slit* and *robo1* gene dosage by half in *slit, robo1/+* embryos results in a mild phenotype where ectopic midline crossing of ap axons is seen in approximately 40% of segments (*Figure 1N, Q*). Homozygous *cyfip* mutants in this sensitized background show a strong enhancement of the ectopic ap crossing defects with 85% of segments exhibiting ectopic crossing (*Figure 1O*). We also visualized all CNS axons using HRP and observed abnormal thickening and fusion of the commissures, a phenotype that bears strong resemblance to *robo1* mutants. Importantly, ap-specific expression of *UAS-CYFIP* significantly rescues the ectopic ap crossing defects but not the pan-neuronal HRP defects (*Figure 1P, Q*) providing strong support for a cell-autonomous role for CYFIP in Slit-Robo1 signaling. Together, these genetic data suggest that the WRC functions in the Slit-Robo1 pathway at the Drosophila midline.

Previous work has identified Rac1 as an important effector of Robo1 signaling in both Drosophila and mouse (*Fan et al., 2003*; *Wong et al., 2001*). SOS is a Rac-GEF that activates Rac1 downstream of Robo1 and is required for Robo1-mediated midline repulsion (*Chance and Bashaw, 2015*; *Yang and Bashaw, 2006*). Since Rac1 is a well-known activator of the WRC (*Chen et al., 2017*; *Chen et al., 2010*; *Eden et al., 2002*; *Ismail et al., 2009*), we reasoned that Rac1 might be responsible for activating the WRC downstream of Robo1, and that Rac1 and the WRC would function cooperatively in the same pathway to regulate Robo1-mediated repulsion. Thus, we predicted that the simultaneous reduction of CYFIP and the Rac-GEF, SOS, would greatly impair Robo1-mediated repulsion, resulting in axons ectopically crossing the midline. As SOS is also maternally deposited (*Yang and Bashaw, 2006*), zygotic *sos* mutants show very mild ectopic crossing defects in approximately 15% of segments (*Figure 2A*). In contrast, double mutants for *sos* and *cyfip* show a striking phenotype in which FasII-positive axons ectopically cross the midline in over 80% of segments (*Figure 2B, C*), a phenotype that bears strong resemblance to the *robo1* mutant phenotype. In addition to examining the phenotype with FasII immunostaining, we also visualized all CNS axons using HRP and observed frequent thickening and fusion of the anterior and posterior commissures, which again bears strong resemblance to *robo1* mutants (*Figure 2B*). Thus, *cyfip* genetically interacts with *sos* to give a strong ectopic crossing phenotype very similar to that seen in *robo1* mutants, supporting the idea that Rac1 and the WRC act cooperatively to regulate midline repulsion.

In Drosophila embryos, both Robo1 and, to a lesser extent, Robo2 contribute to midline repulsion in response to Slit (*Rajagopalan et al., 2000*; *Simpson et al., 2000*). Indeed, on their own *robo2* mutants exhibit only mild phenotypes; however, *robo1, robo2* double mutants exhibit a complete collapse of all CNS axons at the midline, phenocopying the *slit* mutant phenotype. Therefore, mutations in genes that contribute to *robo1* repulsion would be expected to strongly enhance the mild phenotype observed in *robo2* mutants. In *robo2* mutant embryos, FasII-positive axons ectopically

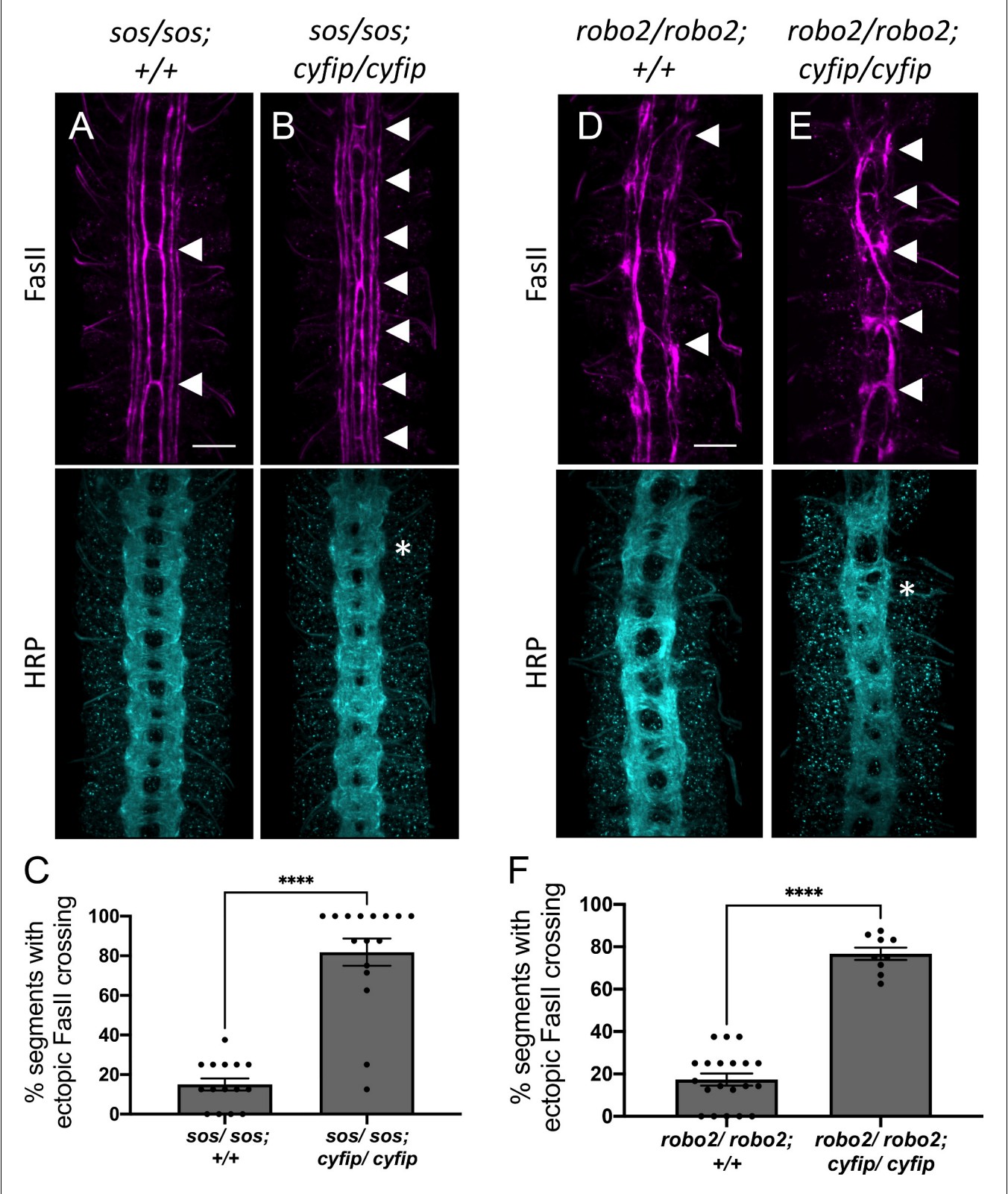

**Figure 2.** The *wave regulatory complex* genetically interacts with *sos* and *robo2*. (A, B, D, E) Stage 17 embryos stained with anti-FasII and anti-HRP. (A) *Sos* embryos show mild ectopic crossing defects of 15% in FasII axons (arrowheads) and no phenotype in HRP. (B) Simultaneous removal of *sos* and *cyfip* results in a very strong enhancement of the ectopic FasII crossing defects to 82% and a strong HRP phenotype with thickening and fusion of commissures (asterisk). Similarly, (D) *robo2* mutants show mild ectopic crossing defects of 17% in FasII axons and a mildly disorganized axon scaffold in

*Figure 2 continued on next page*

*Figure 2 continued*

HRP while (**E**) double mutants for *robo2* and *cyfip* show strong ectopic FasII crossing defects of 77% and thickening and fusion of commissures in HRP. (**C, F**) Quantitation shows the percentage of segments in which FasII axons ectopically cross the midline. Data are presented as mean ± SEM, number of embryos, n = 15 and 16 (for **E**) and 20 and 9 (for **F**). Significance was assessed using Student's *t*-test. Scale bars in (**A**) and (**D**) represent 20 μm.

cross the midline in approximately 17% of segments (*Figure 2D*). In *robo2, cyfip* double mutant embryos, ectopic crossing defects are greatly increased to approximately 75% of segments (*Figure 2E, F*) and axon commissures are thicker and frequently fused, providing additional support for a role for the WRC in midline repulsion. Taken together, these genetic interaction results strongly suggest that the WRC functions in Slit-Robo1-mediated repulsive signaling at the midline.

## The WIRS motif in Robo1 is important for its interaction with the WRC

The cytoplasmic tail of Robo1 contains a WIRS motif, which is conserved in vertebrates (*Figure 3A*). The purified cytoplasmic tail of human Robo1 directly interacts with the WRC in pulldown assays via its WIRS motif (*Chen et al., 2014a*). To determine if this WIRS-dependent interaction with the WRC is conserved in Drosophila Robo1, we performed co-immunoprecipitation assays in Drosophila embryonic S2R+ cells (DGRC, #150) using tagged constructs of Robo1 and HSPC300. The relatively small size of HSPC300 facilitated consistent levels of expression and reduced trial-to-trial variability. We found that Robo1 immunoprecipitated with HSPC300, indicating that Drosophila Robo1, like human Robo1, can also interact with the WRC (*Figure 3C*). Next, we introduced point mutations into the WIRS motif of Robo1 (Robo1ΔWIRS; *Figure 3B*) and found a significant decrease in the amount of Robo1 that immunoprecipitated with HSPC300 (*Figure 3C, E*). Thus, mutating the WIRS motif substantially disrupts the binding of Robo1 to the WRC, indicating that Robo1 interacts with the WRC partly via the WIRS motif. In contrast, the previously published interaction data for human Robo1 (*Chen et al., 2014a*) showed that mutating the WIRS motif completely abolishes binding to the WRC. We speculate that there may be a small amount of indirect binding of Robo1 to the WRC via Ena or DOCK, which are known interactors of Robo1 (*Bashaw et al., 2000*; *Fan et al., 2003*). Previous work has identified interactions between Ena and Abi (*Chen et al., 2014b*) and between the DOCK homolog Nck and Nap1 (*Kitamura et al., 1996*). Both Abi and Nap1 are members of the WRC. As the pulldown assay with human Robo1 was done using purified proteins, any indirect binding will not be detected. Support for this notion comes from our co-immunoprecipitation results of Robo2 and HSPC300. Drosophila Robo2 is structurally similar to Robo1 except that it lacks the CC motifs CC2 and CC3 present in Robo1 that serve as the interaction sites for Ena and DOCK (*Bashaw et al., 2000*; *Fan et al., 2003*; *Figure 3—figure supplement 1A*). Indeed, we find that Robo2 can also interact with HSPC300 though mutating the WIRS motif of Robo2 almost completely abolishes this interaction (*Figure 3—figure supplement 1B, C*). This result is consistent with the idea that there might be indirect binding of the WRC to Robo1 via its interaction with other WRC partners but not to Robo2 that lacks any such interactions.

Next, we wanted to test whether the Robo1-WRC interaction is regulated by the Robo ligand Slit. We treated S2R+ cells with bath application of Slit-conditioned media (Slit-CM) and found a substantial increase in the interaction between Robo1 and HSPC300 as compared to cells treated with mock-CM (*Figure 3D, F*). By contrast, Robo1ΔWIRS shows no significant increase in binding to HSPC300 upon Slit-CM treatment. As there is significant variability in the activity of Slit-CM with each preparation, we see different levels of enhancement in binding obtained with each Slit treatment. Nevertheless, Slit application consistently increases the interaction between Robo1 and HSPC300. These results suggest that upon Slit binding the WRC is recruited to Robo1 via its WIRS motif.

Finally, to test whether this interaction occurs in vivo, we performed co-immunoprecipitation assays using Drosophila embryonic protein lysates. We generated transgenic flies using the GFP-tagged HSPC300 construct and HA-tagged Robo1 constructs. The pan-neuronal *elav-Gal4* driver was used to drive expression of *UAS-HSPC300-GFP* either alone or with the wild-type *UAS-HA-Robo1* or *UAS-HA-Robo1ΔWIRS* transgenes in Drosophila embryos. While wild-type Robo1 co-immunoprecipitates with HSPC300, mutating the WIRS motif results in a significant decrease in this binding (*Figure 3G, H*). These results indicate that Robo1 interacts with the WRC in vivo as well and that this interaction is partly dependent on the WIRS motif.

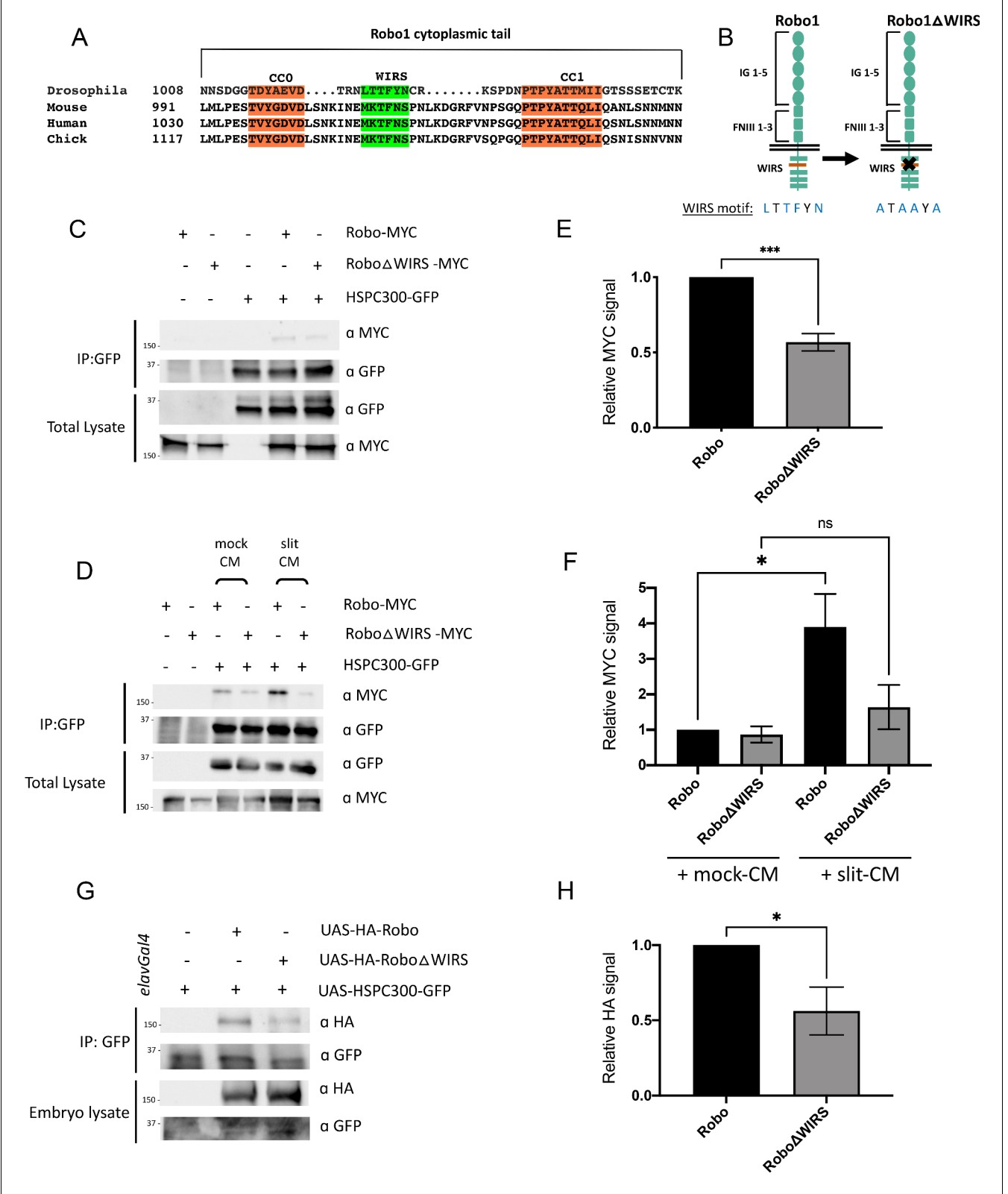

**Figure 3.** Slit-dependent recruitment of the WAVE regulatory complex (WRC) to Robo1 requires the WRC-interacting receptor sequence (WIRS) motif. (**A**) Sequence alignments of the cytoplasmic tail of Robo1 showing that the WIRS motif is conserved across species. (**B**) Schematic depicting the residues of the WIRS motif that are mutated in the Robo1ΔWIRS variant. (**C**) Drosophila S2R+ cell lysates co-expressing HSPC300-GFP with either wild-type Robo1-MYC or Robo1ΔWIRS-MYC were immunoprecipitated with an anti-GFP antibody. The first three lanes show the individual proteins

*Figure 3 continued on next page*

*Figure 3 continued*

expressed alone. The fourth lane shows wild-type Robo1 co-immunoprecipitating with HSPC300 while the fifth lane shows that mutating the WIRS motif decreases this binding. (D) Cell lysates were immunoprecipitated with anti-GFP following a 12 min bath application of mock conditioned media or conditioned media obtained from Slit-expressing cells. The interaction between wild-type Robo1 and HSPC300 is increased in the presence of Slit; however, no significant increase is noted with Robo1ΔWIRS. (E, F) Quantitation of band intensities of the MYC-tagged Robo1 variants in the immunoprecipitates normalized to wild-type Robo1-MYC. Data were normalized to lysate levels of the Robo1 variants and HSPC300 levels in the immunoprecipitates. Error bars represent SEM. Number of trials, n = 4. Significance was assessed using Student's *t*-test (for **E**) and one-way ANOVA with Tukey's multiple comparisons test (for **F**). (G) Lysates from Drosophila embryos with *elavGal4* pan-neuronally driving the expression of HSPC300-GFP alone (lane 1), with wild-type HA-Robo1 (lane 2) or with HA-Robo1ΔWIRS (lane 3), were immunoprecipitated with anti-GFP. Wild-type Robo1 co-immunoprecipitates with HSPC300 and mutating the WIRS motif decreases this binding. (H) Quantitation of band intensities of the HA-tagged Robo1 variants in the imunnoprecipitates normalized to wild-type HA-Robo1. Data were normalized to the lysate levels of the Robo1 variants and HSPC300 levels in the immunoprecipitates. Error bars represent SEM. Number of trials, n = 5. Significance was assessed using Student's *t*-test. Normalized values for the co-immunoprecipitation data are provided in *Figure 3—source data 1*.

The online version of this article includes the following source data and figure supplement(s) for figure 3:

**Source data 1.** Normalized values of co-immunoprecipitation data.
**Figure supplement 1.** Robo2 interaction with the WAVE regulatory complex (WRC) is entirely dependent on its WRC-interacting receptor sequence (WIRS) motif.

## The WIRS motif is essential for Robo1 function in vivo

To test whether this interaction with the WRC is required for Robo1 function in vivo, we compared the gain-of-function and rescue phenotypes of wild-type Robo1 and Robo1ΔWIRS in specific neuronal subsets in the Drosophila ventral nerve cord. We generated transgenic flies with wild-type *UAS-Robo1* or *UAS-Robo1ΔWIRS* constructs. Both the transgenes are tagged with an HA epitope and inserted into the same genomic locus. Immunostaining for HA shows that both transgenes are expressed at comparable levels (*Figure 4D, E*). Using the *eg-Gal4* driver, we expressed these transgenes in eagle neurons, a subset of commissural neurons. Eagle neurons, visualized here using a GFP reporter, consist of two populations: the EG population, which extends its axons in the anterior commissure of a segment, and the EW population, which extends axons in the posterior commissure (*Figure 4A*). Overexpression of wild-type Robo1 in these neurons causes ectopic repulsion from the midline, resulting in a strong gain-of-function phenotype where almost all EW axons fail to cross the midline (*Figure 4B*). In contrast, overexpression of Robo1ΔWIRS results in a significantly weaker gain-of-function phenotype where EW axons in approximately 70% of segments fail to cross the midline (*Figure 4C, F*). Thus, mutating the WRC interaction site on Robo1 hampers its ability to induce ectopic repulsion in vivo.

Next, we assessed the ability of Robo1ΔWIRS to rescue the ectopic crossing defects of FasII-positive axons seen in *robo1* mutant embryos. Unlike in wild-type embryos, where FasII axons never cross the midline (*Figure 4G*), in *robo1* mutants, axons in the medial most fascicle freely cross and recross the midline in 100% of segments (*Figure 4H*). Re-expressing wild-type Robo1 with the pan-neuronal driver *elav-Gal4* restores the ipsilateral projection pattern in most of the segments, lowering the frequency of ectopic crossing to 25% of segments (*Figure 4I*). In contrast, re-expression of Robo1ΔWIRS fails to rescue the crossing defects in 70% of segments (*Figure 4J, K*). This indicates that in the absence of a functional WIRS motif Robo1 is not nearly as effective at restoring repulsive signaling in ipsilateral axons in vivo. Altogether, these results suggest a role for the WIRS motif in Robo1 repulsive signaling at the midline.

## Mutating the endogenous WIRS motif disrupts Robo1 function in vivo

Our in vivo results obtained so far have relied on misexpression or overexpression of Robo1 that likely is not subject to the adequate spatial and temporal regulation that is critical for guidance receptor function. Further, such unregulated high levels of Robo1 expression on the cell surface could potentially mask dysfunction in receptor activity. We see this especially for the rescue experiments with our *UAS-Robo1* transgenes. While the difference in rescue activity between *5XUAS-Robo1* and *5XUAS-Robo1ΔWIRS* is around 50% (*Figure 4K*), performing this rescue assay with *10XUAS-Robo1* and *10XUAS-Robo1ΔWIRS* transgenes, which have double the number of UAS enhancer sites and express much higher levels of the Robo1 variants, gives a much more modest difference of 13% (*Figure 4—figure supplement 1A–E*). Indeed, in rescue experiments using *10XUAS-*

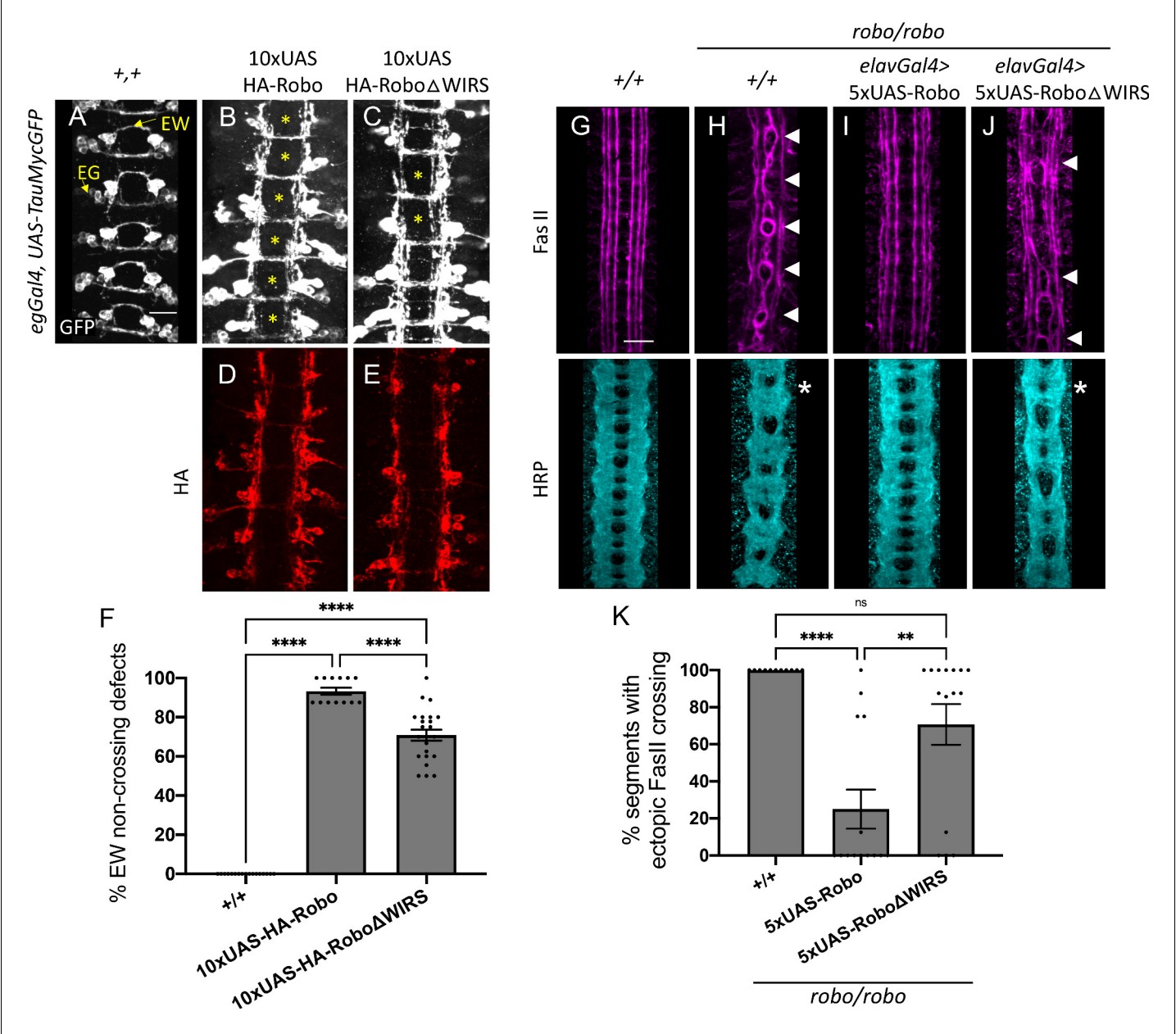

**Figure 4.** The WRC-interacting receptor sequence (WIRS) motif is essential for Robo1 function in vivo. (**A–C**) Stage 16 Drosophila embryos carrying *egGal4* and *UAS-TauMycGFP* transgenes stained with anti-GFP, which labels cell bodies and axons of the eagle neurons (EG and EW). EG neurons project through the anterior commissure of each segment while EW neurons project through the posterior commissure. (**A**) EW neurons cross in 100% of segments in wild-type embryos. (**B**) Misexpression of wild-type HA-tagged Robo1 in eagle neurons results in a strong disruption of midline crossing where EW axons fail to cross in almost all segments of the nerve cord (93%; asterisk). (**C**) Misexpressing HA-tagged Robo1ΔWIRS results in a significantly milder disruption with fewer segments showing EW non-crossing defects (71%). (**D, E**) Embryos stained with anti-HA show comparable expression of the HA-tagged Robo1 variants that were inserted into the same genomic locus. (**F**) Quantitation shows the percentage of segments in which EW axons fail to cross the midline. Data are presented as mean ± SEM, number of embryos, n = 17, 13, 23. Significance was assessed using one-way ANOVA with Tukey's multiple comparisons test. (**G–J**) Stage 17 embryos stained with anti-FasII and anti-HRP. (**G**) Wild-type embryos show no ectopic FasII crossing defects and no phenotype in HRP. (**H**) *Robo* mutants show severe ectopic FasII crossing defects in 100% of segments (arrowheads) and a strong HRP phenotype with thickening and fusion of commissures (asterisk). (**I**) Pan-neuronal expression of wild-type *5XUAS-Robo1* significantly rescues the *robo* mutant phenotype in FasII (to 25%) as well as HRP; however, (**J**) *5XUAS-Robo1ΔWIRS* fails to rescue the *robo* mutant phenotype as efficiently as wild-type Robo1 with frequent ectopic crossing in FasII (71%) and thickened commissures in HRP still evident in these embryos. (**K**) Quantitation shows the percentage of segments in which FasII axons ectopically cross the midline. Data are presented as mean ± SEM, number of embryos, n = 11, 14, 15. Significance was assessed using one-way ANOVA with Tukey's multiple comparisons test. Scale bars in (**A**) and (**G**) represent 20 μm.

*Figure 4 continued on next page*

*Figure 4 continued*

The online version of this article includes the following figure supplement(s) for figure 4:

**Figure supplement 1.** *10XUAS-Robo1* rescue of the *robo* mutant phenotype.

*Robo1* transgenes, we see strong gain-of-function effects that lead to both rescue of abnormal crossing of FasII-positive axons, as well as ectopic repulsion of commissural axons (***Figure 4—figure supplement 1F–J***). Notably, the ectopic repulsion of commissural axons induced by the *10XUAS-Robo1ΔWIRS* transgene is significantly weaker than the ectopic repulsion induced by the wild-type receptor (***Figure 4—figure supplement 1H–J***). Given these caveats, we sought to analyze the function of the WIRS motif in Robo1 signaling in a more endogenous context. First, we performed a rescue assay with an HA-tagged genomic rescue construct of *robo1* that contains upstream and downstream regulatory regions of Robo1 in addition to the Robo1 coding sequence (***Brown et al., 2015***). Transgenics created with this construct show a Robo1 expression pattern that closely resembles that of endogenous Robo1 (***Brown et al., 2015***). We mutated the WIRS motif in this *robo1* genomic rescue construct and inserted the transgene into the same genomic site as the wild-type construct. Both transgenes show comparable levels of Robo1 expression upon HA immunostaining (***Figure 5—figure supplement 1A, B***). We tested the ability of these transgenes to rescue the *robo1* mutant phenotype in FasII-positive axons (***Figure 5B***). One copy of the wild-type *robo1* genomic rescue construct (genRobo) was able to rescue ectopic crossing of FasII-positive axons in almost all segments with only 6% still showing defects (***Figure 5C***) while *robo1ΔWIRS* genomic rescue construct (genRoboΔWIRS) was unable to rescue ectopic crossing defects in over 70% of segments (***Figure 5D, E***). Similarly, for HRP stained axons, the frequent thickening and fusion of the anterior and posterior commissures in *robo1* mutants (***Figure 5B***) can be rescued with the wild-type genRobo but not with genRoboΔWIRS (***Figure 5C, D***). These results, in more physiologically relevant contexts, demonstrate a marked decline in Robo1 function upon disruption of the WRC binding site.

Finally, using the CRISPR-Cas9 system, we mutated the WIRS motif in the endogenous *robo1* locus. We used a single-guide RNA that targets the endogenous WIRS motif and a single-stranded oligonucleotide template to introduce point mutations in the WIRS motif (***Figure 5—figure supplement 2A***). We sequenced the regions surrounding the WIRS motif to verify that we had successfully mutated the WIRS motif without introducing any unwanted frameshift mutations or deletions. While we found no frameshifts, we did notice that our strategy had resulted in an unexplained loss of the smaller intron 16 (***Figure 5—figure supplement 2A***). Since the genRobo constructs and the previously used *robo* swap alleles (***Spitzweck et al., 2010***) that can restore Robo1 function fully do not contain any intronic sequences, we believe that it is extremely unlikely that the loss of this intron affects Robo1 function. Next, we analyzed the phenotypes of both HRP and FasII-positive axons in these *roboΔWIRS* CRISPR embryos. We see a surprisingly strong ectopic crossing phenotype in these embryos with defects in almost 100% of segments, showing that they fully phenocopy the *robo* mutant embryos (***Figure 5B, F***). We were able to achieve a near perfect rescue with the introduction of one copy of genRobo, indicating that this phenotype is not a result of any off-target effects arising from Cas9-mediated cleavage (***Figure 5G, H***). This result also supports our interpretation that the loss of intron 16 in our CRISPR allele has no effect on Robo1 function since the genRobo construct does not include any introns. As an additional control, we also tested whether the *roboΔWIRS* CRISPR mutations disrupt normal Robo1 expression. To investigate this, we immunostained for Robo1 expression using a monoclonal Robo1 antibody. Unlike the *robo* mutants in which no Robo1 protein can be detected (***Figure 5—figure supplement 2C, F***), we see substantial Robo1 staining in the *roboΔWIRS* CRISPR mutants, suggesting that the phenotype is not due to a failure in protein production (***Figure 5—figure supplement 2D, G***). Unlike in wild-type embryos where Robo1 expression is seen primarily on longitudinal tracts and is downregulated on commissures (***Figure 5—figure supplement 2B, E***), in the *roboΔWIRS* CRISPR mutant embryos, we see Robo1 also being expressed on commissures (***Figure 5—figure supplement 2D, G***). While interesting, this observation is not necessarily surprising to us as this altered Robo1 localization on commissures has also been noted in previous studies when Robo1 signaling is disrupted (***Coleman et al., 2010***). Altogether, our genomic Robo rescue assays and *roboΔWIRS* CRISPR mutant phenotypes strongly suggest an important role for the WIRS motif in Robo1 repulsive function in vivo.

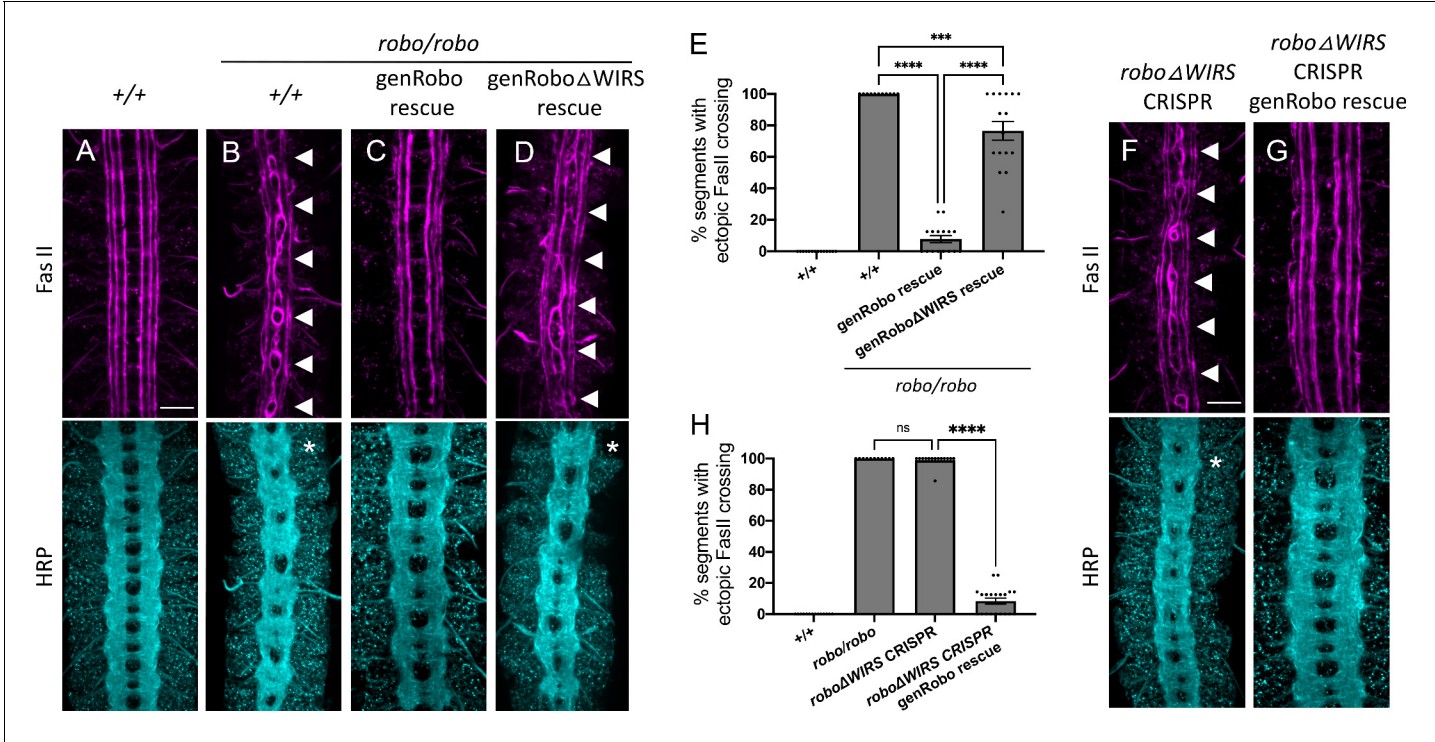

**Figure 5.** Mutating the endogenous WRC-interacting receptor sequence (WIRS) motif disrupts Robo1 function in vivo. (A–D) Stage 17 embryos stained with anti-FasII and anti-HRP. (A) Wild-type embryos showing no phenotype in FasII or HRP. (B) *Robo* mutants show severe ectopic FasII crossing defects in 100% of segments (arrowheads) and a strong HRP phenotype with thickening and fusion of commissures (asterisk). (C) The strong FasII and HRP phenotypes seen in *robo* mutant embryos can be completely rescued with one copy of a wild-type genomic Robo1 rescue construct (genRobo) that contains additional upstream and downstream regulatory regions of *robo1*, more closely mimicking the endogenous Robo1 expression pattern (8%). (D) In contrast, the genomic Robo1 rescue construct containing mutations in the WIRS motif of Robo1 (genRoboΔWIRS) fails to rescue the *robo* mutant phenotype in both FasII (77%) and HRP. (E) Quantitation shows the percentage of segments in which FasII axons ectopically cross the midline. Data are presented as mean ± SEM, number of embryos, n = 14, 11, 16, 16. Significance was assessed using one-way ANOVA with Tukey's multiple comparisons test. (F, G) Stage 17 embryos stained with anti-FasII and anti-HRP. (F) CRISPR embryos with mutations in the endogenous WIRS motif of robo1 show severe phenotypes in FasII and HRP bearing strong resemblance to *robo* mutants. (G) The phenotypes seen in these CRISPR roboΔWIRS embryos can be completely rescued with one copy of the wild-type genomic Robo1 rescue construct (8%). (H) Quantitation shows the percentage of segments in which FasII axons ectopically cross the midline. Data are presented as mean ± SEM, number of embryos, n = 14, 11, 14, 20. Significance was assessed using one-way ANOVA with Tukey's multiple comparisons test. Scale bars in (A) and (F) represent 20 µm.

The online version of this article includes the following figure supplement(s) for figure 5:

**Figure supplement 1.** Comparable expression of the genomic rescue transgenes.

**Figure supplement 2.** Schematic for CRISPR-Cas9 mutagenesis and Robo1 staining in CRIPR roboΔWIRS embryos.

## The Arp2/3 complex interacts genetically and physically with the Slit-Robo pathway

We have shown that the WRC is an important component of the Slit-Robo1 repulsive pathway at the midline. But what happens after the WRC is recruited to Robo1? Is the WRC acting via Arp2/3 to promote branched actin polymerization downstream of Robo1? To address this question, we tested for genetic interactions between *arpc2*, a member of the Arp2/3 complex and the Slit-Robo pathway. Similar to members of the WRC, *arpc2* mutants alone have no ectopic crossing phenotype in FasII axons; however, when introduced into the *slit, robo/+* sensitized background, *arpc2* homozygous mutants show a significant enhancement of the ectopic FasII crossing defects (*Figure 6A–C*), suggesting that the Arp2/3 complex functions in the Slit-Robo repulsive pathway. Additionally, when we remove one copy of *arpc2* together with one copy of *cyfip*, we again observe a significant enhancement of the *slit, robo/+* ectopic crossing defects (*Figure 6—figure supplement 1A–C*). This genetic interaction between *arpc2* and *cyfip* suggests a cooperative effect of the WRC and Arp2/3 in the Slit-Robo1 signaling pathway at the midline. Next, we overexpressed Robo1 in eagle neurons,

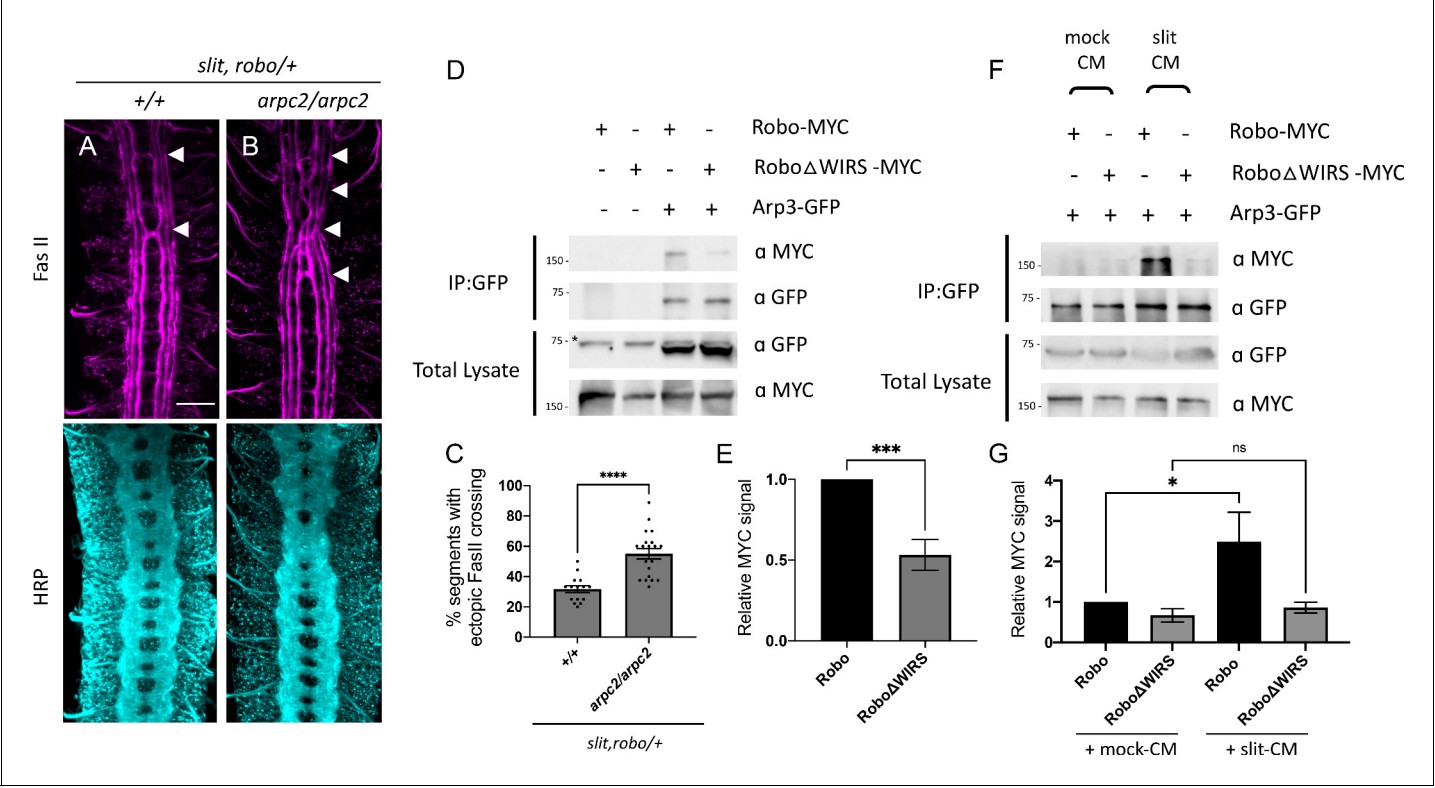

**Figure 6.** The Arp2/3 complex interacts genetically and physically with the Slit-Robo pathway. (**A**, **B**) Stage 17 Drosophila embryos stained with anti-FasII and anti-HRP. (**A**) *Slit, robo* transheterozygous embryos show a mild loss-of-repulsion phenotype with ectopic FasII crossing in 31% of nerve cord segments (arrowheads). (**B**) *Arpc2* homozygous mutants that are simultaneously heterozygous for *slit* and *robo* show ectopic FasII crossing in significantly more segments of the nerve cord (55%). (**C**) Quantitation shows the percentage of segments in which FasII axons ectopically cross the midline. Data are presented as mean ± SEM, number of embryos, n = 15 and 20. Significance was assessed using Student's *t*-test. Scale bar in (**A**) represents 20 μm. (**D**) Drosophila S2R+ cell lysates co-expressing Arp3-GFP with either wild-type Robo1-MYC or Robo1ΔWIRS-MYC were immunoprecipitated with an anti-GFP antibody. The first two lanes show the individual Robo1 variants expressed alone. The third lane shows wild-type Robo1 co-immunoprecipitating with Arp3 while the fourth lane shows that mutating the WIRS motif decreases this binding. Asterisk indicates non-specific bands. (**F**) Cell lysates were immunoprecipitated with anti-GFP following a 12 min bath application of mock conditioned media or conditioned media obtained from Slit-expressing cells. The interaction between wild-type Robo1 and Arp3 is increased in the presence of Slit; however, no significant increase is noted with Robo1ΔWIRS. (**E**, **G**) Quantitation of band intensities of the MYC-tagged Robo1 variants in the immunoprecipitates normalized to wild-type Robo1-MYC. Data were normalized to lysate levels of the Robo1 variants and Arp3 levels in the immunoprecipitates. Error bars represent SEM. Number of trials, n = 7. Significance was assessed using Student's *t*-test (for **E**) and one-way ANOVA with Tukey's multiple comparisons test (for **G**). Normalized values for the co-immunoprecipitation data are provided in *Figure 6—source data 1*.

The online version of this article includes the following source data and figure supplement(s) for figure 6:

**Source data 1.** Normalized values of co-immunoprecipitation data.

**Figure supplement 1.** *arpc2* mutants genetically interact with the Slit-Robo pathway.

**Figure supplement 2.** Comparable surface expression of the wild-type and WRC-interacting receptor sequence (WIRS) mutant forms of Robo1.

which results in a strong gain-of-function phenotype where almost all EW neurons fail to cross the midline. In contrast, overexpressing Robo1 in *arpc2* mutants results in a small but significant suppression of this phenotype (*Figure 6—figure supplement 1D–F*) that is similar to the suppression seen in *cyfip* mutants (*Figure 6—figure supplement 1G–I*), demonstrating a reduction in Robo1's ability to induce ectopic repulsion. Together, these genetic data strongly suggest that the Arp2/3 complex functions in the Slit-Robo1 repulsive pathway.

To determine whether the Arp2/3 complex can physically interact with Robo, we performed co-immunoprecipitation assays in Drosophila embryonic S2R+ cells using tagged constructs of Robo1 and Arp3, another component of the Arp2/3 complex. We found that Robo immunoprecipitated with Arp3, suggesting that the Arp2/3 complex can physically interact with Robo (*Figure 6D*). We

reasoned that if the Arp2/3 complex was being recruited by the WRC to Robo, we would expect that mutating the WIRS motif would disrupt this interaction between Arp2/3 and Robo. Indeed, we found a significant decrease in the amount of RoboΔWIRS that immunoprecipitated with Arp3 as compared to wild-type Robo (*Figure 6D, E*). Furthermore, we can detect an increase in the interaction between Robo and Arp3 in the presence of Slit-CM as compared to mock-CM, suggesting that similar to the WRC, the Arp2/3 complex is also recruited to Robo in response to Slit. By contrast, RoboΔWIRS shows no significant increase in binding to Arp3 in the presence of Slit, demonstrating that the WIRS motif is important for this Slit-dependent recruitment of the Arp2/3 complex to Robo. Together, these observations support the model that upon Slit binding the WRC is recruited to the WIRS motif of Robo and activated, which is in turn responsible for the recruitment of the Arp2/3 complex to facilitate cytoskeletal remodeling downstream of Robo. One possible outcome of initiating localized actin polymerization is the endocytosis and recycling of transmembrane receptors. Indeed, both Drosophila Robo as well vertebrate Robo1 have been previously shown to undergo endocytosis following Slit stimulation (*Chance and Bashaw, 2015*; *Kinoshita-Kawada et al., 2019*). Furthermore, the WRC has been shown to play a role in initiating receptor endocytosis (*Basquin et al., 2015*; *Xu et al., 2016*). Thus, to further evaluate the mechanism of WRC function in Slit-Robo signaling, we investigated whether mutating the WIRS motif in Robo could disrupt signaling by preventing Robo endocytosis. To address this question, we tested whether RoboΔWIRS displays increased surface localization compared to wild-type Robo in both Drosophila embryonic neurons and in cultured dorsal commissural neurons from mice. First, we tested whether Drosophila embryos expressing the genomic HA-tagged Robo rescue transgenes display any difference in surface localization. We dissected embryos live and visualized surface expression of Robo by staining the N-terminal HA tag before fixation and permeabilization (*Figure 6—figure supplement 2A*). Surface Robo was quantified as the mean fluorescence intensity of HA normalized to HRP. We observed no significant difference in the surface expression of Robo and RoboΔWIRS (*Figure 6—figure supplement 2B*). We next cultured E12 mouse dorsal commissural neurons that were electroporated with either wild-type MYC-tagged human Robo1 (hRobo1) or MYC-tagged hRobo1ΔWIRS. Following a 30 min bath application with Slit, we visualized surface expression of hRobo1 by staining the N-terminal MYC tag before fixation and permeabilization (*Figure 6—figure supplement 2C*). Surface hRobo1 was quantified as the mean fluorescence intensity of MYC, and the analysis was limited to Robo3-positive commissural neurons. Here again, we observed no significant difference in the surface localization of hRobo1 and hRobo1ΔWIRS (*Figure 6—figure supplement 2D*), suggesting that the WIRS motif has no detectable effect on Robo1 surface levels. Together, these observations point to a non-endocytic role for the WRC in promoting Robo repulsion.

## The WIRS motif is required for Slit-dependent repulsion in mouse spinal commissural axons

The WIRS motif in the Robo1 receptor is conserved in vertebrates, raising the possibility for a potential role in vertebrate Robo1 signaling. Indeed, the cytoplasmic tail of human Robo1 can bind to the WRC via its WIRS motif (*Chen et al., 2014a*). Thus, to address the question of whether the WIRS motif is important for vertebrate Robo1 signaling, we introduced point mutations into the WIRS motif of hRobo1 and performed gain-of-function experiments with wild-type hRobo1 and hRobo1ΔWIRS constructs. We electroporated E12 mouse spinal cords with wild-type hRobo1 or hRobo1ΔWIRS, along with RFP as a reporter for efficiency of electroporation and cultured dorsal spinal cord explants next to mock 293 T cell aggregates or cell aggregates expressing Slit (*Figure 7A*). We observe poor penetration of the anti-MYC antibody in explants and hence use RFP as a measure of electroporation efficiency. We observe comparable levels of RFP staining in explants (*Figure 7—figure supplement 1A*). Explants cultured adjacent to mock cell aggregates show uniform outgrowth on all sides of the explant (*Figure 7B*). In contrast, explants cultured adjacent to Slit-expressing aggregates show decreased outgrowth on the side proximal to the Slit-expressing aggregate as compared to the distal side (*Figure 7C*). Explants electroporated with wild-type hRobo1 show an increased repulsive response to Slit with even less outgrowth on the proximal side and a significantly lower proximal/distal outgrowth ratio (*Figure 7D, F*). In contrast, explants electroporated with hRobo1ΔWIRS show no such gain-of-function response to Slit and have a proximal/distal outgrowth ratio similar to that seen for RFP electroporated explants (*Figure 7E, F*), suggesting that the WIRS motif is important for the Slit-induced repulsive response of vertebrate Robo1. Next, to assess whether

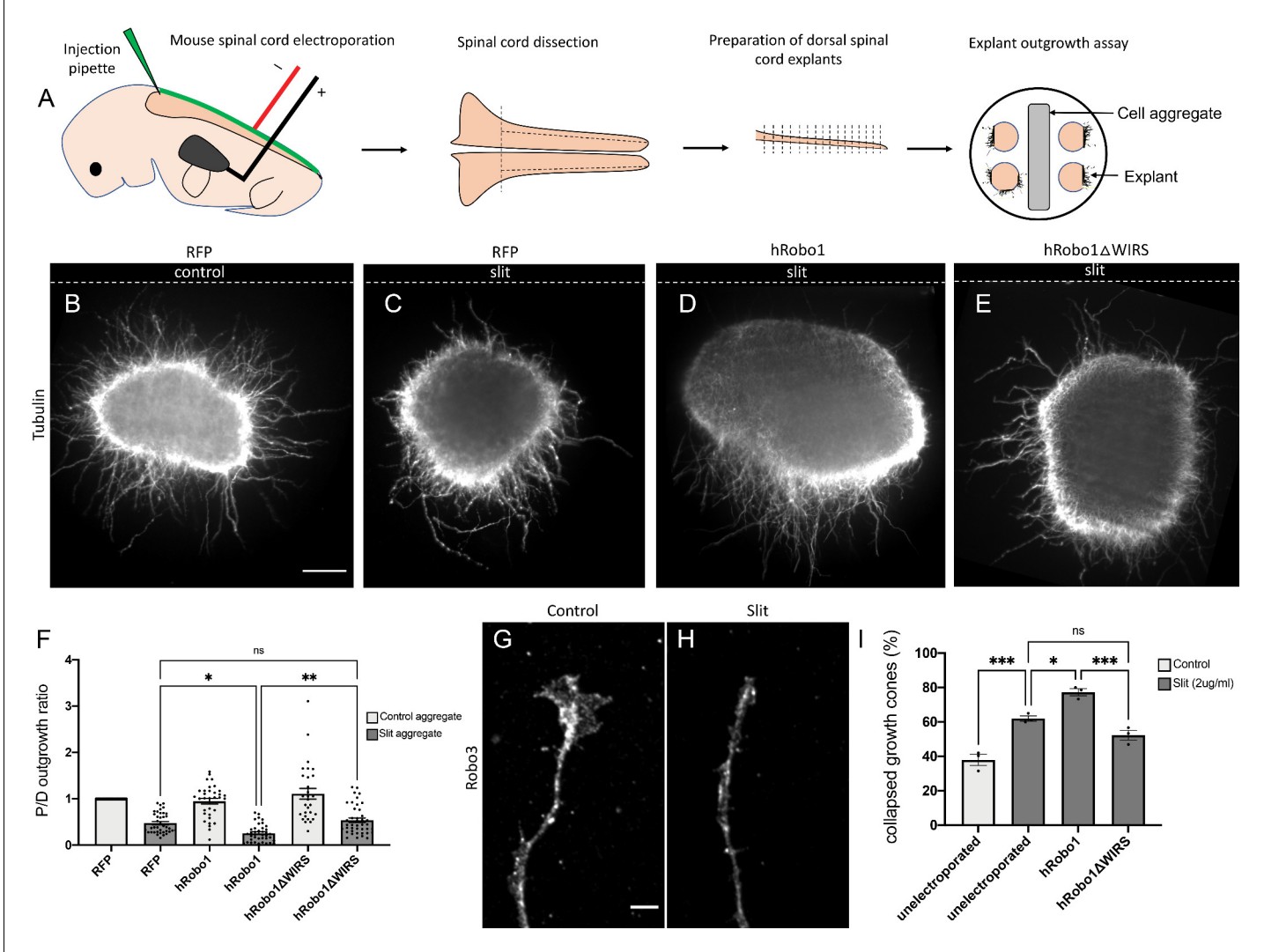

**Figure 7.** The WRC-interacting receptor sequence (WIRS) motif is required for Slit-dependent Robo1 repulsion in mouse spinal commissural axons. (**A**) Schematic of electroporation and culture of spinal cord explants. Dotted lines show cut sites to obtain dorsal spinal cord explants. The image on the right depicts the arrangement of explants cultured around a 293 T cell aggregate (control or Slit-expressing) embedded in collagen. (**B–E**) E12 dorsal spinal cord explants labeled with anti-tubulin to visualize axon outgrowth. Dotted lines indicate the position of the cell aggregate. (**B**) RFP electroporated explant cultured next to a mock cell aggregate shows uniform outgrowth on all sides of the explant. (**C**) RFP electroporated explant cultured next to a Slit-expressing cell aggregate shows decreased outgrowth on the quadrant proximal to the aggregate as compared to the quadrant distal to it (0.47). (**D**) Explant electroporated with wild-type hRobo1 cultured next to a Slit-expressing cell aggregate shows even less outgrowth on the proximal quadrant demonstrating increased responsiveness to Slit (0.14). (**E**) Explant electroporated with hRobo1ΔWIRS cultured next to a Slit-expressing cell aggregate shows no such increase in Slit responsiveness as the proximal: distal outgrowth ratio is similar to that seen for RFP electroporated explants (0.54). (**F**) Quantification shows the proximal:distal outgrowth ratio for explants cultured next to control cell aggregates (white) and Slit-expressing cell aggregates (gray). Data are presented as mean ± SEM, number of explants, n = 29, 39, 33, 39, 29, 41 (from three independent experiments). Significance was assessed using one-way ANOVA with Tukey's multiple comparisons test. (**G, H**) Growth cone collapse in response to Slit in E12-dissociated commissural axons. Growth cone morphology was examined by staining for the commissural marker Robo3. (**I**) Quantification shows percentage of axons with collapsed growth cones. Unelectroporated neurons show an increased level of collapse when treated with Slit (from 38% without Slit to 62% with bath application of Slit). Neurons electroporated with wild-type hRobo1 show a gain-of-function response to Slit with an even higher collapse level (77%). In contrast, neurons electroporated with hRobo1ΔWIRS show no gain-of-function and a collapse level similar to unelectroporated neurons (52%). For neurons electroporated with the MYC-tagged hRobo1 variants, only Robo3- and MYC-positive axons were analyzed. Data are presented as mean ± SEM, number of trials, n = 3 (over 30 neurons for each condition/trial). Significance was assessed using one-way ANOVA with Tukey's multiple comparisons test. Scale bars represent 100 μm in (**B**) and 5 μm in (**G**).

The online version of this article includes the following figure supplement(s) for figure 7:

**Figure supplement 1.** Expression of hRobo1 variants electroporated into dorsal spinal commissural neurons.

the WIRS motif is also important for the collapsing activity of Robo1 in response to Slit, we performed Slit-induced collapse assays using dissociated E12 mouse dorsal spinal commissural neurons (*Figure 7G, H*). In our control cultures, 38% of Robo3-positive commissural axons show collapsed growth cones (*Figure 7I*). Following a 30 min treatment with recombinant Slit2, we see an increase in the collapse rate to 62%. Neurons electroporated with wild-type MYC-tagged hRobo1 show a further increase in collapse rate with 77% of Robo3- and MYC-positive axons ending in collapsed growth cones. In contrast, we saw no increase in the number of collapsed growth cones in neurons electroporated with MYC-tagged hRobo1ΔWIRS (*Figure 7I*), suggesting that the WIRS motif is also important for the Slit-induced collapsing activity of Robo1. The hRobo1 variants show comparable levels of MYC staining in neurons (*Figure 7—figure supplement 1B, C*).

To study the function of the Robo1 WIRS motif in an in vivo context, we examined its role in commissural axon guidance in the embryonic chicken spinal cord. We reasoned that unilateral expression of Robo1 in pre-crossing commissural neurons would prevent their axons from crossing the floor plate by inducing a premature responsiveness to midline-secreted Slits (*Brose et al., 1999*; *Long et al., 2004*). To do this, we used in ovo electroporation to introduce a GFP expression plasmid either alone (Control) or with MYC-tagged wild-type human Robo1 or human Robo1ΔWIRS expression constructs into pre-crossing commissural neurons at Hamburger–Hamilton (HH) stage 14 (*Hamburger and Hamilton, 1951*). At HH stage 22–23, a 'crossing index' was calculated by measuring GFP and MYC signal in the contralateral side of the spinal cord as a fraction of GFP and MYC signal on the electroporated side (*Figure 8D*). We found that ectopic expression of wild-type Robo1 and GFP resulted in a GFP crossing index of 0.21 ± 0.13% (mean ± SD, n = 6), which was significantly less than that of GFP alone (Control), with a crossing index of 1.8 ± 1.1% (n = 6, p=0.004), indicating that Robo1 expression was sufficient to block commissural crossing (*Figure 8A, B, E*). Robo1ΔWIRS and GFP overexpression resulted in a GFP crossing index of 0.68 ± 0.60% (n = 8), which was not significantly different from that of wild-type Robo1 (p=0.472; *Figure 8C, E*). However, quantification of the crossing index based on the MYC tag fused to the wild-type Robo1 and Robo1ΔWIRS constructs resulted in a significantly higher MYC crossing index of Robo1ΔWIRS-expressing neurons (1.7 ± 0.97%, n = 8) compared to that of wild-type Robo1-expressing neurons (0.53 ± 0.36%, n = 6, p=0.013; *Figure 8F*). The disparity between the effects of the WIRS mutation calculated using GFP and MYC-based quantification may reflect a greater efficiency of GFP plasmid transduction and expression compared to the Robo1 expression constructs. These data demonstrate a significant reduction in Robo1's ability to prevent spinal commissural crossing in the absence of the WIRS motif.

Altogether, the results from mouse dorsal spinal cord explants and dissociated neuron cultures along with the in vivo experiments in chick embryos show that while overexpression of wild-type hRobo1 is able to enhance the repulsive response to Slit, mutating the WIRS motif in hRobo1 abolishes this gain-of-function response. These observations indicate that the WIRS motif is important for vertebrate Robo1 signaling and suggest an evolutionarily conserved role for the WIRS motif in Robo1 repulsive signaling.

## Discussion

In this article, we have documented a conserved role for the WRC in Slit-mediated Robo1 repulsive signaling. Using the developing Drosophila embryonic CNS, we demonstrate a series of dose-dependent genetic interactions between components of the WRC and Slit-Robo1 signaling, which show that the WRC functions in vivo to regulate Robo1 repulsive signaling at the midline. Biochemical experiments in cultured cells show that Robo1 can bind to the WRC partially via its WIRS motif and that Slit stimulation can induce recruitment of the WRC to Robo1. Further, we present several lines of evidence to demonstrate that the WIRS motif is important for Robo1 function in vivo. First, mutating the WIRS motif results in a significantly weaker gain-of-function phenotype when Robo1 is misexpressed in commissural axons. Second, the Robo1 variant with mutations in its WIRS motif fails to rescue the *robo1* mutant phenotype as effectively as wild-type Robo1. Finally, mutating the WIRS motif in the endogenous *robo1* locus using the CRISPR-Cas9 system results in embryos with severe ectopic crossing defects that phenocopy *robo1* mutants. These data demonstrate a severe decline in Robo1 function upon disruption of the WRC binding site. Together, our observations support the model that Slit stimulation results in recruitment of the WRC to the WIRS motif in Robo1, which is

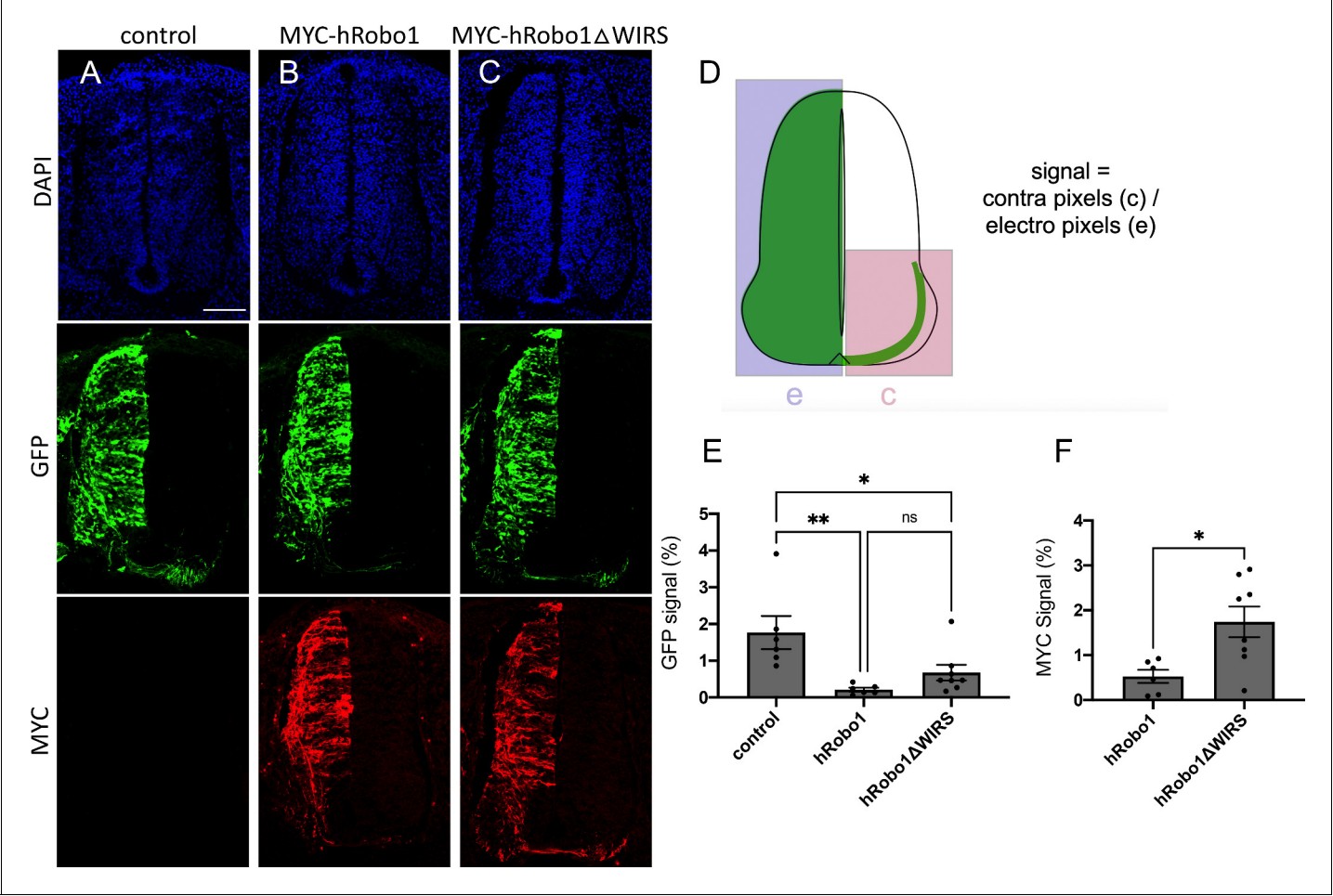

**Figure 8.** The WRC-interacting receptor sequence (WIRS) motif is required for vertebrate Robo1-mediated ectopic repulsion in vivo. (A–C) Transverse sections of Hamburger–Hamilton (HH) stage 22–23 chicken spinal cords electroporated with GFP alone or together with MYC-tagged hRobo1 or hRobo1ΔWIRS and stained with DAPI, anti-GFP, and anti-MYC. (A) Electroporation of GFP alone shows numerous GFP-positive axons crossing the midline with a GFP crossing index of 1.8%. (B) Electroporation of GFP together with MYC-hRobo1 shows far fewer GFP-positive axons crossing the midline and very few MYC-positive axons on the contralateral side, with GFP and MYC crossing indices of 0.21 and 0.53%, respectively. (C) In contrast, electroporation of GFP along with MYC-hRobo1ΔWIRS shows substantially more GFP- and MYC-positive axons crossing the midline with higher GFP and MYC crossing indices of 0.68 and 1.7%, respectively. (D) Crossing index (signal) is the GFP or MYC fluorescence signal in the contralateral side of the spinal cord expressed as a fraction of total GFP or MYC fluorescence signal in the electroporated side. (E, F) Quantitation of crossing index for GFP and MYC signal. Data are presented as mean ± SEM, number of embryos, n = 6, 6, 8 (for E) and 6, 8 (for F). Significance was assessed using one-way ANOVA with Tukey's multiple comparisons test (for E) and Student's *t*-test (for F). Scale bar represents 100 µm in (A).

vital to Robo1-mediated repulsive signaling at the midline (*Figure 9*). Further, using genetic and biochemical approaches, we show that the Arp2/3 complex functions in the Slit-Robo signaling pathway and undergoes a WIRS-dependent recruitment to Robo1 in response to Slit. We propose that downstream of Robo1 the WRC functions to recruit the Arp2/3 complex to initiate localized cytoskeletal remodeling. We also present several lines of evidence that support an evolutionarily conserved role for the WIRS motif in vertebrate Robo1 signaling. First, we show that Robo1ΔWIRS is less effective at mediating repulsion in response to Slit in explants from the mouse dorsal spinal cord. In addition, Robo1ΔWIRS is less responsive to the collapsing activity of Slit in dissociated spinal commissural axons. Finally, we show that mutating the WIRS motif in human Robo1 results in a reduced ability to induce ectopic repulsion in embryonic chicken commissural axons as compared to wild-type human Robo1. These data highlight a vital conserved role for the WIRS motif in Robo1 function.

In this study, we used a series of complementary approaches to evaluate the importance of the WIRS motif for Robo1 repulsive signaling. While it is generally assumed that the high expression

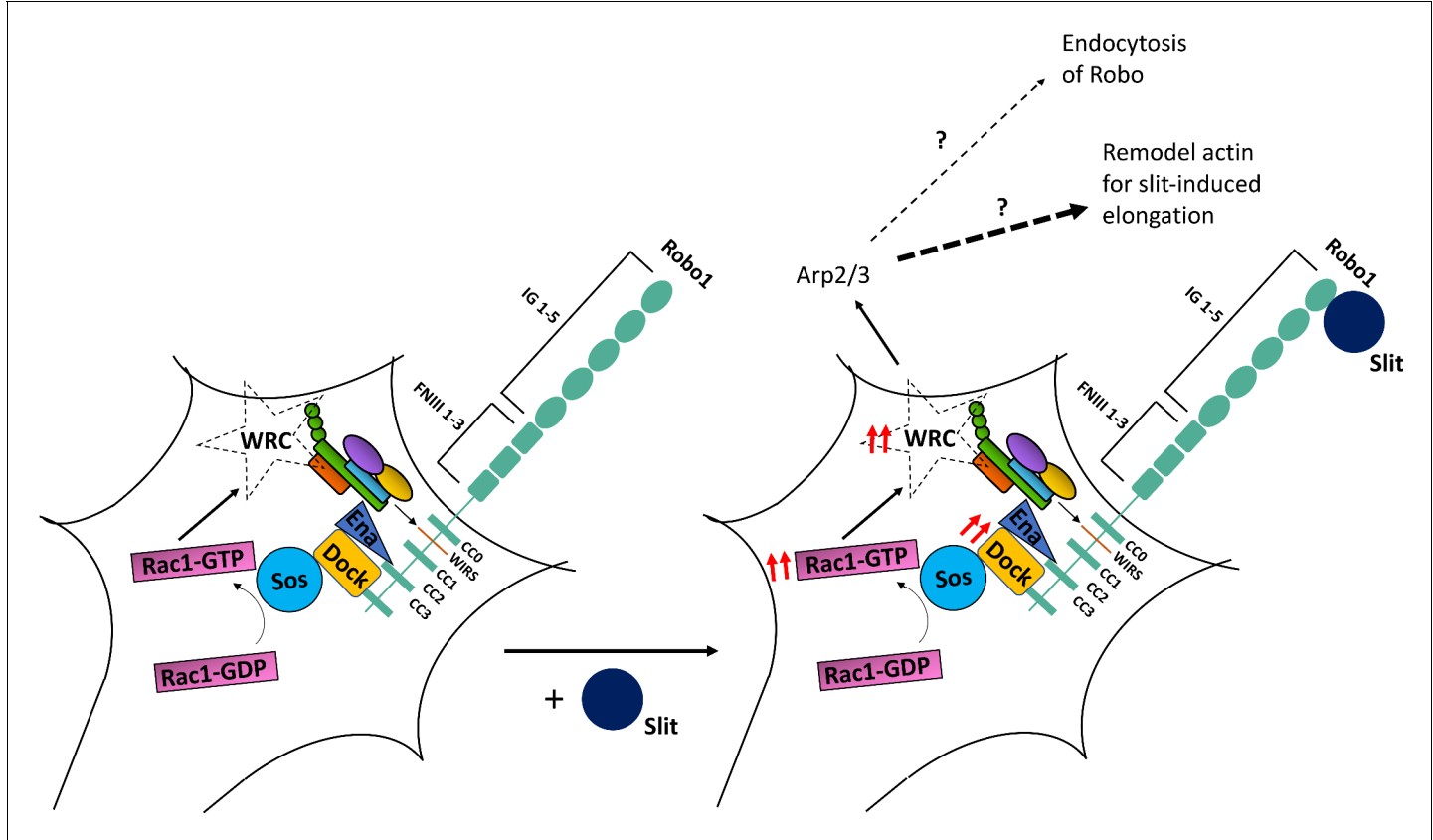

**Figure 9.** A model of WAVE regulatory complex (WRC) function in Robo1 signaling. In our proposed model, the WRC binds to Robo1 partly via its WRC-interacting receptor sequence (WIRS) motif. Rac1 is activated downstream of Robo1 (*Fan et al., 2003*; *Wong et al., 2001*), which likely activates the complex. Slit binding induces increased WIRS-dependent recruitment of the WRC to Robo1, which is vital to Robo1-mediated repulsive signaling. WRC functions downstream of Robo1 by activating Arp2/3 to remodel the actin cytoskeleton. We hypothesize that these WIRS-WRC-mediated actin rearrangements are more likely to facilitate an initial extension of Slit-induced filopodia than endocytosis or recycling of the Robo1 receptor.

levels resulting from the Gal4/UAS system are unlikely to reflect normal spatial and temporal regulation, it remains unclear to what extent this might confound comparisons between different mutant variants of a given protein. Our results with Robo1 indicate that Gal4-UAS/directed expression significantly hinders the detection of critical structural elements of the receptor. For example, when using a pan-neuronal driver to reintroduce Robo1 into the *robo1* mutant embryos, we observe only very modest differences between the wild-type and WIRS mutant forms of overexpressed Robo1. In contrast, we see much more severe phenotypes when the WIRS motif is disrupted under conditions that more closely match the endogenous robo1 levels using the *robo1* genomic rescue constructs or when mutating the WIRS motif in the endogenous robo1 locus using the CRISPR-Cas9 system. These direct comparisons between different assays to measure protein function suggest that rescue experiments using the Gal4/UAS system must be interpreted with caution. This also suggests that our results comparing the gain-of-function effects of Robo1 and Robo1ΔWIRS in vertebrate systems are likely to underestimate the significance of the WIRS motif and recruitment of the WRC for Robo1 repulsion.

Here, we have shown that the WRC is an important component of the Slit-Robo1 repulsive pathway at the midline. But what happens once the WRC is recruited to Robo1? The VCA region of Scar/ Wave is sequestered within the complex until activation, which triggers a conformational change releasing the VCA domain (*Chen et al., 2010*; *Ismail et al., 2009*). Rac1 is an important activator of the WRC and has been previously found to be activated downstream of Robo1 in both Drosophila and mouse (*Fan et al., 2003*; *Wong et al., 2001*). The genetic interaction between *sos* and *cyfip* suggests that these proteins function cooperatively to regulate midline repulsion. We propose a

model where the WRC is recruited to the WIRS motif in response to Slit where it is activated by increased local Rac1 signaling. Active WRC can then promote Arp2/3-mediated actin assembly. Such a direct interaction with the WRC via the WIRS motif would allow for localized WRC activity at desired subdomains to achieve tighter spatiotemporal control and support directional actin changes. At first glance, the initiation of actin polymerization downstream of Robo1 might seem paradoxical; however, many repulsive guidance cues recruit downstream effectors that enhance actin polymerization. One recent study demonstrates that dorsal root ganglion neurons initially extend filopodia toward a source of Slit before retracting (*McConnell et al., 2016*). The McConnell study highlights the nuanced and complex actin rearrangements that occur downstream of guidance cues, which potentially contribute to sensing of the environment for improved resolution of a guidance gradient. The WRC might play an important role in the generation of these Slit-induced filopodia by initiating the formation of branched actin filaments that are subsequently rebundled to form filopodia as suggested by the convergent elongation model that supports a role for Arp2/3 in filopodia formation in neurons (*Yang and Svitkina, 2011*). Ena/VASP proteins, downstream effectors of Robo1, are important for these Slit-induced filopodial extensions (*McConnell et al., 2016*) and on account of their actin bundling activity are perfectly poised to orchestrate this actin reorganization in order to drive filopodia formation. The WRC has also been shown to be important for receptor endocytosis (*Basquin et al., 2015*; *Xu et al., 2016*), and another possible outcome of initiating localized actin polymerization is the endocytosis and recycling of Robo1. Indeed, previous work has demonstrated that endocytosis of Drosophila Robo1 upon Slit stimulation is essential for Robo1 repulsive signaling (*Chance and Bashaw, 2015*) and that vertebrate Robo1 also undergoes endocytosis and recycling following Slit stimulation (*Kinoshita-Kawada et al., 2019*), suggesting a conservation of this regulatory mechanism. However, our data suggests that mutating the WIRS motif has no detectable effect on the surface localization of Robo1 either in Drosophila or in cultured mouse commissural neurons. Future studies are needed to decipher the exact nature and function of the local actin remodeling induced by the Rac-WRC-Arp2/3 complex signaling axis downstream of Robo1. Additionally, other known downstream effectors of Robo1 like Ena and Abl have also been shown to influence WRC activity (*Chen et al., 2014b*; *Leng et al., 2005*). It would thus be interesting to dissect how this inter-regulation between these different components of Robo1 signaling contributes to fine-tuning of WRC activity to generate a specific output for Slit-Robo1 repulsion.

Previously, the WIRS motif has been shown to be important in Neuroligins and Syg-1 for proper synapse formation (*Chia et al., 2014*; *Xing et al., 2018*) as well as in Neogenin for the maintenance of adherens junctions (*Lee et al., 2016*). In these contexts, it is apparent that the WRC reinforces the F-actin network at these membrane junctions. However, it is unclear if the WIRS-WRC interactions are subject to regulation by the respective ligands or if the WRC performs more of a scaffolding function. In the context of axon guidance, our work demonstrates a ligand-dependent recruitment of the WRC to the WIRS domain of Robo1 suggestive of both spatial and temporal specificity of WRC activation. To our knowledge, this study is the first to demonstrate that the WRC can be recruited to a guidance receptor in response to a ligand.

In addition to functioning as part of the actin-regulating complex, WRC members can have functions independent of the complex as well. For example, CYFIP proteins can interact with fragile X mental retardation protein (FMRP) to regulate mRNA localization and protein translation (*Abekhoukh and Bardoni, 2014*; *Schenck et al., 2003*; *Schenck et al., 2001*) and Abi can interact with WASP and Diaphanous to regulate F-actin (*Bogdan et al., 2005*; *Ryu et al., 2009*). While we cannot entirely rule out a role for WRC-independent functions of these proteins in Robo1 signaling, several lines of evidence point to the involvement of the WRC as a whole downstream of Robo1. First, two separate WRC members, *cyfip* and *hscp300*, show genetic interactions with the Slit-Robo pathway. Second, the physical interaction between Robo1 and HSPC300 is dependent on the WIRS motif, which requires a binding interface formed by CYFIP and Abi that comes together only in the fully assembled WRC (*Chen et al., 2014a*). Additionally, the strong midline crossing phenotypes we see upon manipulating the WIRS motif suggests that it is indeed this interaction with the fully assembled WRC that is important for Robo1 signaling in vivo. Finally, we tested whether the Drosophila homolog of FMRP, d*fmr1*, genetically interacts with the Slit-Robo1 pathway. In contrast to *cyfip* and *hspc300*, completely removing d*fmr1* has no effect on the *slit, robo* transheterozygous phenotype (*Figure 6—figure supplement 1J*). This data further supports a WRC-dependent function for *cyfip*

in Slit-Robo signaling and suggests that CYFIP/dFMR interactions are not important for Robo repulsion.

Drosophila Robo1 has numerous functions in development outside of its role in midline repulsion, and the *robo1ΔWIRS* CRISPR mutants generated here also provide an opportunity to discern which developmental functions of Robo1 require the WRC in future studies. Robo1 regulates the migration of chordotonal sensory neurons (**Kraut and Zinn, 2004**) and mesodermal migration for muscle patterning (**Kramer et al., 2001**). Embryos lacking *robo1* show defects in heart lumen formation (**Qian et al., 2005**) and tracheal migration (**Englund et al., 2002**). In mammals, Robo1 also plays important roles outside of the nervous system, including the formation of blood vessels (**Rama et al., 2015**) and organs like the heart (**Mommersteeg et al., 2013**) and the mammary glands (**Macias et al., 2011**), and it can also regulate stem cell proliferation (**Ballard et al., 2015**). Finally, the Slit-Robo pathway has been shown to regulate tumor angiogenesis along with tumor cell migration and metastasis (**Tong et al., 2019**). Mis-regulation of Slit-Robo signaling has been implicated in multiple types of tumorigenesis, making it a promising target for cancer treatments (**Koohini et al., 2019**). Such therapeutic avenues require a comprehensive understanding of Slit-Robo signaling in specific cancers, highlighting the importance of investigating the WRC as a downstream effector of Robo in disease contexts as well.

In addition to Robo1, other Robo receptors also contain WIRS motifs in their cytoplasmic domains. Drosophila Robo2 plays a minor role in midline repulsion and, together with Robo3, also regulates lateral positioning of the longitudinal fascicles (**Evans and Bashaw, 2010**; **Rajagopalan et al., 2000**). As Robo2 and Robo3 do not contain CC2 and CC3 domains, to which most of the known Robo1 effectors bind, very little is known about their downstream signaling. Vertebrate Robo3 can induce repulsive signaling in response to a recently identified ligand, NELL2 (**Jaworski et al., 2015**) Vertebrate Robo3 also contains a WIRS motif in its cytoplasmic domain, raising the possibility that these Robo receptors could share a common cellular mechanism for repulsion, despite responding to distinct ligands. Additionally, attractive axon guidance receptors like Fra and its vertebrate ortholog DCC also contain WIRS motifs in their cytoplasmic domains. Unsurprisingly, many core actin-modifying proteins that act downstream of repulsive cues like Ena/VASP and Abl kinase also function in attractive signaling (**Forsthoefel et al., 2005**; **Gitai et al., 2003**). We can speculate that the WRC might also function in both repulsion and attraction by regulating different actin-based processes like membrane trafficking versus growth cone advancement. Alternatively, as other studies have shown the importance of an initial growth cone extension toward repulsive cues, it is likely that tight spatiotemporal activation along with regulation by other effector molecules can result in fine-tuning of WRC activity to contribute to distinct cytoskeletal outputs downstream of different guidance receptors.

## Materials and methods

### Key resources table

| Reagent type (species) or resource | Designation | Source or reference | Identifiers | Additional information |
|---|---|---|---|---|
| Cell line (*Homo sapiens*) | 293T | ATCC | ATCC CRL-3216 | RRID:CVCL_0063 Authenticated via STR profiling using ATCC services |
| Cell line (*Drosophila melanogaster*) | S2R+ | Drosophila Genomics Resource Center | Cat#150 | RRID:CVCL_Z831 Authenticated by morphology and doubling time |
| Genetic reagent (*Mus musculus*) | CD-1 line | Charles River | Stock#022 | RRID:IMSR_CRL:022 |
| Genetic reagent (*D. melanogaster*) | *D. melanogaster: w¹¹¹⁸* | **Chance and Bashaw, 2015** | N/A | |
| Genetic reagent (*D. melanogaster*) | *D. melanogaster: robo^GA285* | **Chance and Bashaw, 2015** | N/A | |

*Continued on next page*

*Continued*

| Reagent type (species) or resource | Designation | Source or reference | Identifiers | Additional information |
|---|---|---|---|---|
| Genetic reagent (*D. melanogaster*) | *D. melanogaster: slit²* | *Chance and Bashaw, 2015* | N/A | |
| Genetic reagent (*D. melanogaster*) | *D. melanogaster: sos⁴ᴳ* | *Yang and Bashaw, 2006* | N/A | |
| Genetic reagent (*D. melanogaster*) | *D. melanogaster: robo2ˣ¹²³* | *Evans and Bashaw, 2010* | N/A | |
| Genetic reagent (*D. melanogaster*) | *D. melanogaster: scar^{Δ37}* | Bloomington Drosophila Stock Center | BDSC: 8754 | RRID:BDSC_8754 |
| Genetic reagent (*D. melanogaster*) | *D. melanogaster: arpc2^{KG04658}* | Bloomington Drosophila Stock Center | BDSC: 13978 | RRID:BDSC_13978 |
| Genetic reagent (*D. melanogaster*) | *D. melanogaster: hspc300^{Δ54.3}* | Kind gift from A. Giangrande | N/A | |
| Genetic reagent (*D. melanogaster*) | *D. melanogaster: cyfip^{Δ85.1}* | Kind gift from A. Giangrande | N/A | |
| Genetic reagent (*D. melanogaster*) | *D. melanogaster: fmr1³* | Kind gift from T. Jongens | N/A | |
| Genetic reagent (*D. melanogaster*) | *D. melanogaster: apGal4* | *Evans and Bashaw, 2010* | N/A | |
| Genetic reagent (*D. melanogaster*) | *D. melanogaster: egGal4* | *Evans and Bashaw, 2010* | N/A | |
| Genetic reagent (*D. melanogaster*) | *D. melanogaster: UAS-CD8GFP* | *Evans and Bashaw, 2010* | N/A | |
| Genetic reagent (*D. melanogaster*) | *D. melanogaster: UAS-TauMycGFP* | *Evans and Bashaw, 2010* | N/A | |
| Genetic reagent (*D. melanogaster*) | *D. melanogaster: 10XUAS-HA-Robo1 86 F8* | *Evans and Bashaw, 2010* | N/A | |
| Genetic reagent (*D. melanogaster*) | *D. melanogaster: 10XUAS-HA-Robo1ΔWIRS 86 F8* | This paper | N/A | Available from Bashaw lab; methods: genetic stocks |
| Genetic reagent (*D. melanogaster*) | *D. melanogaster: 5XUAS-HA-Robo1 86 F8* | *Chance and Bashaw, 2015* | N/A | |
| Genetic reagent (*D. melanogaster*) | *D. melanogaster: 5XUAS-HA-Robo1ΔWIRS 86 F8* | This paper | N/A | Available from Bashaw Lab; methods: genetic stocks |
| Genetic reagent (*D. melanogaster*) | *D. melanogaster: UAS-CYFIP* | Kind gift from A. Giangrande | N/A | |
| Genetic reagent (*D. melanogaster*) | *D. melanogaster: robo1::HArobo1 28E7* | Kind gift from T. Evans | N/A | |
| Genetic reagent (*D. melanogaster*) | *D. melanogaster: robo1::HArobo1ΔWIRS 28E7* | This paper | N/A | Available from Bashaw lab; methods: genetic stocks |
| Genetic reagent (*D. melanogaster*) | *D. melanogaster: robo1ΔWIRS CRISPR* | This paper | N/A | Available from Bashaw Lab; methods: genetic stocks |
| Recombinant DNA reagent | Plasmid: pCAG-MYC-hRobo1 | This paper | N/A | Available from Bashaw lab; methods: molecular biology |

*Continued on next page*

Continued

| Reagent type (species) or resource | Designation | Source or reference | Identifiers | Additional information |
|---|---|---|---|---|
| Recombinant DNA reagent | Plasmid: pCAG-MYC-hRobo1ΔWIRS | This paper | N/A | Available from Bashaw lab; methods: molecular biology |
| Recombinant DNA reagent | Plasmid: pCAG-RFP | Kind gift from A. Jaworski | N/A | |
| Recombinant DNA reagent | Plasmid: pSec TagB-hSlit2-MYC | Kind gift from A. Chedotal | N/A | |
| Recombinant DNA reagent | Plasmid: p10UAS TattB-HA-Robo1 | *Evans and Bashaw, 2010* | N/A | |
| Recombinant DNA reagent | Plasmid: p10UAS TattB-HA-Robo1ΔWIRS | This paper | N/A | Available from Bashaw Lab; methods: molecular biology |
| Recombinant DNA reagent | Plasmid: p10UAST-HSPC300-GFP | This paper | N/A | Available from Bashaw Lab; methods: molecular biology |
| Recombinant DNA reagent | Plasmid: p5UAST attB-HA-Robo1 | *Chance and Bashaw, 2015* | N/A | |
| Recombinant DNA reagent | Plasmid: p10UAST attB-HA-Robo1ΔWIRS | This paper | N/A | Available from Bashaw Lab; methods: molecular biology |
| Recombinant DNA reagent | Plasmid: pUAST-Slit | *Chance and Bashaw, 2015* | N/A | |
| Recombinant DNA reagent | Plasmid: pMT-Gal4 | *Chance and Bashaw, 2015* | N/A | |
| Recombinant DNA reagent | Plasmid: robo1 genomic rescue construct | Kind gift from T. Evans | N/A | |
| Recombinant DNA reagent | Plasmid: robo1ΔWIRS genomic rescue construct | This paper | N/A | Available from Bashaw Lab; methods: molecular biology |
| Recombinant DNA reagent | Plasmid: pCFD3-dU6:3gRNA | Addgene | Plasmid#49410 | RRID:Addgene_49410 |
| Recombinant DNA reagent | Plasmid: p10UAST attB-Robo1-MYC | This paper | N/A | Available from Bashaw Lab; methods: molecular biology |
| Recombinant DNA reagent | Plasmid: p10UASTattB-Robo1ΔWIRS-MYC | This paper | N/A | Available from Bashaw Lab; methods: molecular biology |
| Recombinant DNA reagent | Plasmid: p10UAST attB-Robo2-MYC | This paper | N/A | Available from Bashaw Lab; methods: molecular biology |
| Recombinant DNA reagent | Plasmid: p10UAST attB-Robo2ΔWIRS-MYC | This paper | N/A | Available from Bashaw Lab; methods: molecular biology |
| Antibody | Mouse monoclonal anti-MYC | DSHB | Cat#9E10-C | IF (1:500), WB (1:1000), RRID:AB_2266850 |
| Antibody | Mouse monoclonal anti-HA | BioLegend | Cat#901502 | IF (1:500), WB (1:1000), RRID:AB_2565007 |
| Antibody | Mouse monoclonal anti-beta tubulin | DSHB | Cat#E7-S | IF (1:300), WB (1:1000), RRID:AB_528499 |

*Continued on next page*

*Continued*

| Reagent type (species) or resource | Designation | Source or reference | Identifiers | Additional information |
|---|---|---|---|---|
| Antibody | Chick polyclonal anti-beta gal | Abcam | Cat#ab9361 | IF (1:500), RRID:AB_307210 |
| Antibody | Mouse monoclonal anti-Fasciclin II | DSHB | 1D4 | IF (1:50), RRID:AB_528235 |
| Antibody | Rabbit polyclonal anti-GFP | Invitrogen | Cat#a11122 | IF (1:250), WB (1:500), IP (1:500), RRID:AB_221569 |
| Antibody | Rabbit polyclonal anti-dsRed | Takara | Cat#632496 | IF (1:200), RRID:AB_10013483 |
| Antibody | Mouse monoclonal anti-Scar (supernatant) | DSHB | Cat#P1C1 | IF (1:50), RRID:AB_2618386 |
| Antibody | Mouse monoclonal anti-Robo (supernatant) | DSHB | Cat#13C9 | IF (1:50), RRID:AB_2181861 |
| Antibody | Mouse monoclonal anti-Slit (supernatant) | DSHB | Cat#C555.6D | WB (1:100), RRID:AB_528470 |
| Antibody | Goat polyclonal anti-Robo3 | R&D Systems | Cat#AF3076 | IF (1:200), RRID:AB_2181865 |
| Antibody | Alexa647 Goat polyclonal anti-HRP | Jackson Immunoresearch | Cat#123-605-021 | IF (1:500), RRID:AB_2338967 |
| Antibody | Goat polyclonal anti-Mouse HRP | Jackson Immunoresearch | Cat#115-035-146 | WB (1:10,000), RRID:AB_2307392 |
| Antibody | Goat polyclonal anti-Rabbit HRP | Jackson Immunoresearch | Cat#111-035-003 | WB (1:10,000), RRID:AB_2313567 |
| Antibody | 488 Donkey polyclonal anti-Mouse | Jackson Immunoresearch | Cat#715-545-150 | IF (1:500), RRID:AB_2340846 |
| Antibody | 488 Donkey polyclonal anti-Goat | Jackson Immunoresearch | Cat#705-165-147 | IF (1:500), RRID:AB_2307351 |
| Antibody | Alexa488 Goat polyclonal anti-Rabbit | Invitrogen | Cat#A11034 | IF (1:500), RRID:AB_2576217 |
| Antibody | Alexa488 Goat polyclonal anti-Mouse | Invitrogen | Cat#A11029 | IF (1:500), RRID:AB_138404 |
| Antibody | Alexa488 Goat polyclonal anti-Chick | Invitrogen | Cat#A11039 | IF (1:500), RRID:AB_142924 |
| Antibody | Cy3 Goat polyclonal anti-Mouse | Jackson Immunoresearch | Cat#115-165-003 | IF (1:500), RRID:AB_2338680 |
| Antibody | Cy3 Goat polyclonal anti-Rabbit | Jackson Immunoresearch | Cat#111-165-003 | IF (1:500), RRID:AB_2338000 |
| Antibody | Cy3 Goat polyclonal anti-Chick | Abcam | Cat#ab97145 | IF (1:500), RRID:AB_10679516 |
| Peptide, recombinant protein | Poly-D-lysine | Sigma Aldrich | Cat#P6407 | |
| Peptide, recombinant protein | N-Cadherin | R&D Systems | Cat#1388-NC | |
| Peptide, recombinant protein | Netrin-1 | R&D Systems | Cat#1109-N1/CF | |

*Continued on next page*

*Continued*

| Reagent type (species) or resource | Designation | Source or reference | Identifiers | Additional information |
|---|---|---|---|---|
| Peptide, recombinant protein | Slit2N | R&D Systems | Cat#5444-SL-050 | |
| Chemical compound, drug | KCl | Thermo Fisher | Cat#BP366-1 | |
| Chemical compound, drug | $MgCl_2$ | Thermo Fisher | Cat#BP214-500 | |
| Chemical compound, drug | HEPES | Thermo Fisher | Cat#BP299-1 | |
| Chemical compound, drug | L15 media | Gibco | Cat#11415-064 | |
| Chemical compound, drug | Horse serum | Gibco | Cat#16050122 | |
| Chemical compound, drug | Fastgreen dye | Thermo Fisher | Cat#F99-10 | |
| Chemical compound, drug | Opti-MEM | Gibco | Cat#31985-070 | |
| Chemical compound, drug | F12 | Gibco | Cat#11765-054 | |
| Chemical compound, drug | Glucose | Thermo Fisher | Cat#D16-500 | |
| Chemical compound, drug | 100x Pen/Strep/Glutamine | Gibco | Cat#10378-016 | |
| Chemical compound, drug | HBSS | Gibco | Cat#14175-079 | |
| Chemical compound, drug | Trypsin | Gibco | Cat#25300054 | |
| Chemical compound, drug | DNAse I | New England Biolabs | Cat#M0303L | |
| Chemical compound, drug | $MgSO_4$ | Thermo Fisher | Cat#7487-88-9 | |
| Chemical compound, drug | Neurobasal | Gibco | Cat#21103-049 | |
| Chemical compound, drug | FBS | Gibco | Cat#10437-028 | |
| Chemical compound, drug | B-27 | Thermo Fisher | Cat#A3582801 | |

*Continued on next page*

*Continued*

| Reagent type (species) or resource | Designation | Source or reference | Identifiers | Additional information |
|---|---|---|---|---|
| Chemical compound, drug | Effectene Transfection Reagent | Qiagen | Cat#301425 | |
| Chemical compound, drug | Rat Tail Collagen | Corning | Cat#354249 | |
| Chemical compound, drug | Paraformaldehyde 16% solution, EM grade | Electron Microscopy Services | Cat#15710 | |
| Chemical compound, drug | Amicon Ultracel 30K filters | Millipore | Cat#UFC903096 | |
| Chemical compound, drug | Drosophila Schneider's Media | Life Technologies | Cat#21720024 | |
| Chemical compound, drug | Surfact-AMPS NP40 | Thermo Fisher | Cat#85124 | |
| Chemical compound, drug | Protease Inhibitor (Complete) | Roche | Cat#11697498001 | |
| Chemical compound, drug | 2x Laemmli Sample Buffer | Bio-Rad | Cat#1610737 | |
| Chemical compound, drug | 4x Laemmli Sample Buffer | Bio-Rad | Cat#1610747 | |
| Chemical compound, drug | Clarity Western ECL Substrate | Bio-Rad | Cat#1705061 | |
| Chemical compound, drug | *Not*I | New England Biolabs | Cat#R3189S | |
| Chemical compound, drug | *Xba*I | New England Biolabs | Cat#R0145S | |
| Chemical compound, drug | *Bgl*II | New England Biolabs | Cat#R0144S | |
| Chemical compound, drug | *Xho*I | New England Biolabs | Cat#R0146S | |
| Chemical compound, drug | *Bbs*I | New England Biolabs | Cat#R3539S | |
| Chemical compound, drug | *Mfe*I | New England Biolabs | Cat#R3589S | |
| Chemical compound, drug | Proteinase K | Roche Diagnostics | Cat#03115828001 | |
| Chemical compound, drug | Protein A Agarose beads | Invitrogen | Cat#15918-014 | |
| Chemical compound, drug | rProteinG Agarose beads | Invitrogen | Cat#15920–010 | |
| Commercial assay or kit | Quikchange II site-directed mutagenesis kit | Agilent | Cat#200523 | |
| Software, algorithm | Image J Fiji | Fiji | https://imagej.net/Fiji | RRID:SCR_002285 |
| Software, algorithm | Adobe Photoshop | Adobe | N/A | RRID:SCR_014199 |
| Software, algorithm | Bio-Rad Image Lab | Bio-Rad | http://www.bio-rad.com/zh-cn/product/image-lab-software | RRID:SCR_014210 |

*Continued on next page*

*Continued*

| Reagent type (species) or resource | Designation | Source or reference | Identifiers | Additional information |
|---|---|---|---|---|
| Software, algorithm | GraphPad Prism 9 | GraphPad software | https://www.graphpad.com/ | RRID:SCR_002798 |
| Software, algorithm | ChemiDoc Imaging System | Bio-Rad | Cat#171001401 | |
| Software, algorithm | Volocity Software | Perkin Elmer | http://cellularimaging.perkinelmer.com/downloads/ | RRID:SCR_002668 |
| Other | BTX Electroporator | BTX Harvard Apparatus | Cat#45-0662 | |
| Other | Nikon Ti-U microscope | Nikon | N/A | |

## Contact for reagent and resource sharing

Further information and requests for resources and reagents should be directed to the lead contact, Greg J. Bashaw (gbashaw@pennmedicine.upenn.edu).

## Genetic stocks

The following Drosophila strains were used: $w^{1118}$, $robo^{GA285}$, $slit^2$, $sos^{4G}$, $robo2^{x123}$, $scar^{\Delta37}$, $arpc2^{KG04658}$, *apGal4*, *egGal4*, *UAS-CD8GFP II*, *UAS-TauMycGFP III*, *10XUAS-HA-Robo1 86F8*, and *5XUAS-HA-Robo1 86F8*. Fly strains $hspc300^{\Delta54.3}$, $cyfip^{\Delta85.1}$, and *UAS-CYFIP* were a kind gift from A. Giangrande. The $fmr1^3$ strain was a kind gift from T. Jongens. The genomic *robo1* rescue strain *robo1::HArobo1 28E7* was a kind gift from T. Evans. The following transgenic stocks were generated: *10UAS-HA-RoboΔWIRS 86F8*, *5UAS-HA-RoboΔWIRS 86F8*, *robo1::HArobo1ΔWIRS 28E7*, *10UAS-HSPC300-GFP 86F8*. Transgenic flies were generated by BestGene Inc (Chino Hills, CA) using ΦC31-directed site-specific integration into landing sites at cytological position 86F8 (For *UAS-Robo* constructs) or 28E7 (for genomic *robo1* rescue constructs). Genomic *robo1::HArobo1ΔWIRS 28E7* rescue transgene was introduced onto a $robo^{GA285}$ chromosome via meiotic recombination, and the presence of the $robo^{GA285}$ mutation was confirmed in all recombinant lines by DNA sequencing. The CRISPR line *robo1ΔWIRS* was generated by cloning a guide targeting the WIRS motif into a pCFD3-dU6:3 backbone (Addgene, #49410) and sending positive clones to BestGene Inc for injection. Flies were screened by PCR and restriction digest followed by DNA sequencing. All crosses were carried out at 25℃.

## Mice

Timed pregnant female CD-1 mice were obtained from Charles River. All animal work was approved by the Institutional Animal Care and Use Committee (IACUC) of the University of Pennsylvania. Embryos of both sexes were randomly used for spinal cord explants and primary dissociated neuron cultures.

## Chicken

All animal experiments were carried out in accordance with the Canadian Council on Animal Care guidelines and approved by the IRCM Animal Care Committee and the McGill University Animal Care Committee. Fertilized chicken eggs (FERME GMS, Saint-Liboire, QC, Canada) were incubated (Lyon Technologies, model PRFWD) at 39℃ according to standard protocols.

## Dissociated commissural neuron culture

Primary commissural neuron cultures were prepared as described previously (Langlois 2010) and maintained at 5% $CO_2$ in a humidified incubator. Briefly, commissural neurons were isolated from E12.5 dorsal spinal cords and plated on acid-washed, poly-D-lysine (Sigma, #P6407) and 2 μg/ml N-cadherin (R&D, #1388-NC) coated coverslips. Neurons were cultured in Neurobasal medium supplemented with 10% heat-inactivated FBS (Gibco, #10437-028) and 1X penicillin/streptomycin/

glutamine (Gibco, #10378-016). After ~20 hr, the medium was replaced with Neurobasal supplemented with 1X B-27 (Thermo, #A3582801) and the neurons were used for experiments 1 hr later.

## Explant culture

Dorsal spinal cord explants from E12.5 embryos were dissected and cultured in collagen gels as described previously (Serafini 1994). Briefly, explants were cultured in 50% OptiMEM (Gibco, #31985-070) and 45% Ham's F-12 (Gibco, #11765-054) media supplemented with 5% horse serum (HS, Gibco, #16050122), 0.75% glucose (Thermo, #D16-500), and 1X penicillin/streptomycin/glutamine for 48 hr with 500 ng/ml Netrin-1 (R&D, #1109-N1/CF).

## Cell culture

Drosophila S2R+ cells (DGRC, Cat#150) were maintained at 25°C in Schneider's media (Life Technologies, #21720024) supplemented with 10% (vol/vol) FBS and a mixture of 1% penicillin and streptomycin. Morphology and doubling time were used for validation of the cell line. The cells grow as a loose semi-adherent monolayer with a doubling time of about 40 hr. 293 T cells (ATCC CRL-3216) were maintained at 37°C and 5% $CO_2$ in a humidified incubator in DMEM (Gibco, #11965084) supplemented with 10% (vol/vol) FBS and a mixture of 1% penicillin and streptomycin. Cells were authenticated by STR profiling using ATCC Cell Line Authentication services. Mycoplasma testing was negative for both cell lines.

## Method details

### Molecular biology

For making the *p10UAST-HA-Robo1ΔWIRS, p10UAST-Robo1ΔWIRS-MYC,* and the *p5UAST-HA-Robo1ΔWIRS*constructs, the wild-type Robo1 coding sequences from *p10UAST-HA-Robo1, p10UAST-Robo1-MYC,* and the *p5UAST-HA-Robo1* constructs were subcloned into the smaller pBlueScript backbone and point mutations were introduced into the WIRS motif of the Robo coding sequences with the Quikchange II site-directed mutagenesis kit (Agilent, #200523) using the following primers: GACACCCGTAACGCTACCGCCGCCTACGCTTGTCGCAAG and CTTGCGACAAGCG TAGGCGGTAGCGTTACGGGTGTC. The mutated Robo1 coding sequences were then subcloned back into the respective vectors with 10xUAS or 5xUAS sequences and an attB site for ΦC31-directed site-specific integration. A similar strategy was used for making *p10UAST-Robo2ΔWIRS-MYC* using the following primers: ACCGACTATGCAGAGGCGTCCGCTGCTGGCAAGGCA and TGCCTTGCCAGCAGCGGACGCCTCTGCATAGTCGGT. For the genomic rescue *robo1::HArobo1ΔWIRS* construct, the same Robo1 primers mentioned above were used to mutate the WIRS motif using Quikchange and the mutated Robo1 coding sequence was cloned into the genomic rescue construct backbone (kind gift from T. Evans) using *Bgl*II.

For *MYC-hRobo1ΔWIRS*, the following primers were used for Quikchange: AACAAAATCAA TGAGGCGAAAGCCGCCAATAGCCCAAATCTGAAG and CTTCAGATTTGGGCTATTGGCGGC TTTCGCCTCATTGATTTTGTT. Next, wild-type *MYC-hRobo1* and *MYC-hRobo1ΔWIRS* coding sequences were cloned into a pCAG vector (provided by A. Kania) using *Not*I/*Xho*I sites. A signal peptide sequence was included upstream of the MYC tag.

For making the *p10UAST-HSPC300-GFP* construct, *hspc300* cDNA was PCR amplified from the pOT2 BGDP Gold Collection (clone# FI14118) and tagged with a C-terminal GFP separated by a linker using overlap extension PCR with the following primers: TATATAGCGGCCGCCACCATGAG TGGGGCT and CGCGCGTCTAGATCACTTGTACAGCTCGTC and overlapping primers GG TGAAACATTAACGGGACATATGGGAGGAATGGTGAGCAAGGGC and GCCCTTGCTCACCATTCC TCCCATATGTCCCGTTAATGTTTCACC. This PCR fragment was then cloned into a *p10UAST* plasmid containing an attB site using *Not*I/*Xba*I sites. For making the *p10UAST-Arp3-GFP* construct, *arp3* cDNA was PCR amplified from the pOT2 BGDP Gold Collection (clone# LD35711) and cloned into the *p10UAST-HSPC300-GFP* vector described above by swapping out the *hspc300* insert using *Not*I/*Nde*I sites and the following primers: TATATA-GCGGCCGC-CACC-ATGGCAGGCAGGCTAC and GGTCCATATGTGTCATGGTGCCAAAGACGGGATTGT.

## CRISPR Cas9-mediated mutagenesis

For synthesizing the guide RNA to target the WIRS motif in the endogenous *robo1* locus, the following sense and antisense oligonucleotides were used: GTCGGCGTACGGCGTGGGATTAT and AAACATAATCCCACGCCGTACGC. This guide RNA was selected with zero predicted off-target effects using http://targetfinder.flycrispr.neuro.brown.edu. The oligos were annealed and cloned into a *Bbs*I-digested pCFD3-dU6:3 vector. A single-stranded oligonucleotide template was designed to introduce point mutations into the WIRS motif. These mutations also destroy the gRNA target sequence and the PAM sequence to prevent subsequent cleavages by Cas9. An *Mfe*I site is mutated, which was used for screening potential CRISPR flies. The sequence of the template used is: CAA TCCAACTACAATAACTCCGATGGAGGAACCGATTATGCAGAAGTTGACACCCGTAATGC TACCGCCGCCTACGCTTGTCGCAAGGTGAGGATCATATGAATTGCATCACACAACAATTTC. The template along with the pCFD3 vector containing the guide RNA was sent to BestGene Inc for injection. The progeny from these flies were crossed to balancer stocks to generate stable lines. Flies from these lines were then screened with *Mfe*I following genomic DNA extraction, and positive hits were sent for DNA sequencing.

## Immunoprecipitation

S2R+ cells were transiently transfected with Effectene transfection reagent (Qiagen, Valencia, CA, #301425) and induced 24 hr later with 0.5 mM copper sulfate. 24 hr after induction, cells were lysed in TBS-V (150 mM NaCl, 10 mM Tris pH-8, 1 mM ortho-vanadate) supplemented with 0.5% Surfact-AMPS NP40 (Thermo, Waltham, MA, #85124) and 1x Complete Protease Inhibitor (Roche, #11697498001) for 20 min at 4°C. Soluble proteins were recovered by centrifugation at 15,000 × g for 10 min at 4°C. Lysates were pre-cleared with 30 µl of a 50% slurry of protein A (Invitrogen, #15918-014) and protein G agarose beads (Invitrogen, #15920-010) by incubation for 20 min at 4°C. Pre-cleared lysates were then incubated with 0.7 µg of rabbit anti-GFP antibody for 2 hr at 4°C to precipitate HSPC300-GFP. After incubation, 30 µl of a 50% slurry of protein A and protein G agarose beads was added and samples were incubated for an additional 30 min at 4°C. The immunocomplexes were washed three times with lysis buffer, boiled for 10 min in 2x Laemmli SDS sample buffer (Bio-Rad, #1610737), and analyzed by western blotting. Proteins were resolved by SDS-PAGE and transferred to nitrocellulose membrane (Amersham, #10600032). Membranes were blocked with 5% dry milk and 0.1% Tween 20 in PBS for 1 hr at room temperature and incubated with primary antibodies overnight at 4°C. Following three washes with PBS/0.1% Tween 20, membranes were incubated with the appropriate HRP-conjugated secondary antibody at room temperature for 1 hr. Signals were detected using Clarity ECL (Bio-Rad, #1705061) according to manufacturer's instructions. For preparation of Slit-CM, cells were transfected with a *pUAST-Slit* vector and a *PMT-Gal4* vector using Effectene transfection reagent. Gal4 was induced 24 hr later with 0.5 mM copper sulfate. 24 hr after induction, Slit-CM was collected and concentrated using Amicon filters (Amicon Ultracel 30K, Millipore, #UFC903096). For CM treatment, cells were incubated with control-CM (prepared using an empty *pUAST* vector) or Slit-CM on an orbital shaker at room temperature for 12 min, then lysed for immunoprecipitation as described above. Antibodies used: for immunoprecipitation, rabbit anti-GFP and for western blot, rabbit anti-GFP (1:500, Invitrogen, #a11122), mouse anti-MYC (1:1000, DSHB, #9E10-C), mouse anti-Slit (1:50, DSHB, #C555.6D), HRP goat anti-rabbit (1:10,000, Jackson Immunoresearch, #111-035-003), and HRP goat anti-mouse (1:10,000, Jackson Immunoresearch, #115-035-146).

For co-immunoprecipitation assays in Drosophila embryos, embryonic protein lysates were prepared from approximately 100 µl of embryos overexpressing *UAS-HSPC300-GFP* alone or with the HA-tagged *UAS-Robo1* variants in all neurons. Embryos were lysed in 0.5 ml TBS-V (150 mM NaCl, 10 mM Tris pH 8.0, 1 mM ortho-vanadate) supplemented with 1% Surfact-AMPS NP40 and protease inhibitors by manual homogenization using a plastic pestle. Homogenized samples were incubated with gentle rocking at 4°C for 10 min and centrifuged at 15,000 × g for 10 min in a pre-chilled rotor. Supernatants were collected after centrifugation, and immunoprecipitations and western blotting were performed as described above. Antibodies used: for immunoprecipitation, rabbit anti-GFP (1:500, Invitrogen, #a11122) and for western blot, rabbit anti-GFP (1:500, Invitrogen, #a11122), mouse anti-HA (1:1000, BioLegend, #901502), mouse anti-beta tubulin (1:1000, DSHB, #E7), HRP

goat anti-rabbit (1:10,000, Jackson Immunoresearch, #111-035-003), and HRP goat anti-mouse (1:10,000, Jackson Immunoresearch, #115-035-146).

## Immunostaining

Dechorionated, formaldehyde-fixed Drosophila embryos were fluorescently stained using standard methods. The following antibodies were used: rabbit anti-GFP (1:250, Invitrogen, #a11122), mouse anti-HA (1:500, BioLegend,#901502), chick anti-beta gal (1:500, Abcam, #ab9361), mouse anti-Scar (1:50, DSHB, #P1C1), mouse anti-Robo (1:50, DSHB, #13C9), Alexa647 goat anti-HRP (1:500, Jackson Immunoresearch, #123-605-021), Alexa488 goat anti-rabbit (1:500, Invitrogen, #A11034), Alexa488 goat anti-mouse (1:500, Invitrogen, #A11029), Alexa488 goat anti-chick (1:500, Invitrogen, #A11039), Cy3 goat anti-mouse (1:500, Jackson Immunoresearch, #115-165-003), Cy3 goat anti-Chick (1:500, Abcam, #ab97145), and 647 goat anti-HRP (1:1000, Jackson Immunoresearch, #123-605-021). Embryos were filleted and mounted in 70% glycerol/1× PBS. Surface staining of the HA-tagged genomic Robo rescue transgenes in Drosophila embryos was carried out as previously described (*Bashaw, 2010*). Briefly, embryos were dissected live, blocked with in 5% normal goat serum (NGS) in PBS for 15 min at 4°C, and stained with mouse anti-HA (1:500, BioLegend, #901502) in PBS for 30 min at 4°C. Following washes with PBS, embryos were fixed in 4% paraformaldehyde (Electron Microscopy Services, #15710) for 15 min at 4°C. Following washes with PBS, fixed embryos were then permeabilized with 0.1% Triton X-100 in PBS (PBT) for 10 min and stained with 647 goat anti-HRP (1:1000, Jackson Immunoresearch, #123-605-021) in 5% NGS in PBT overnight at 4°C. Following washes with PBT, secondary antibody consisting of Alexa488 goat anti-mouse (1:500, Invitrogen, #A11029) diluted in 5% NGS in PBT was added and incubated for 1 hr at room temperature. Embryos were then washed with PBT and mounted in Aquamount.

Dissociated spinal commissural neurons were fixed for 20 min in 4% paraformaldehyde (Electron Microscopy Services, #15710) at room temperature and washed three times with PBS. Fixed neurons were then permeabilized with 0.1% Triton X-100 in PBS (PBT) for 10 min and blocked with 2% HS in PBT for 30 min at room temperature. The blocking solution was replaced with primary antibody diluted in 2% HS in PBT and incubated overnight at 4°C. Following three washes with PBT, secondary antibody diluted in 2% HS in PBT was added and incubated for 1 hr at room temperature. Neurons were then washed three times with PBT and the coverslips were mounted in Aquamount. The following antibodies were used: mouse anti-MYC (1:500, DSHB, #9E10-C), goat anti-Robo3 (1:200, R&D Systems, #AF3076), Cy3 donkey anti-goat (1:500, Jackson Immunoresearch, #705-165-147), and 488 donkey anti-goat (1:500, Jackson Immunoresearch, #715-545-150). For surface staining of MYC-tagged hRobo1 variants in dissociated commissural neurons, neurons were treated with recombinant hSlit2-N (R&D, #5444-SL-050) at 2 μg/ml for 30 min at 37°C, following which neurons were blocked in 5% NGS in PBS for 15 min at 4°C and stained with mouse anti-MYC (1:500, DSHB, #9E10-C) in PBS for 30 min at 4°C. Following two washes with PBS, neurons were fixed in 4% paraformaldehyde (Electron Microscopy Services, #15710) for 20 min at 4°C. Following washes with PBS, fixed neurons were then permeabilized with 0.1% Triton X-100 in PBS (PBT) for 10 min and stained with goat anti-Robo3 (1:200, R&D Systems, #AF3076) in 5% NGS in PBT overnight at 4°C. Following washes with PBT, secondary antibody consisting of 488 donkey anti-goat (1:500, Jackson Immunoresearch, #715-545-150) and Cy3 donkey anti-goat (1:500, Jackson Immunoresearch, #705-165-147) diluted in 5% NGS in PBT was added and incubated for 1 hr at room temperature. Neurons were then washed with PBT and mounted in Aquamount.

Collagen-embedded explants were fixed in 4% paraformaldehyde overnight at 4°C and washed three times for 10 min in PBS. Fixed explants were then blocked in 2.5% NGS in PBT for 2 hr at room temperature and incubated with primary antibody diluted in blocking solution overnight at 4°C. Explants were washed six times for 1 hr with PBT and incubated with secondary antibody diluted in blocking solution overnight at 4°C. After six 1 hr washes with PBT, explants were mounted on cavity slides. The following antibodies were used: mouse anti-MYC (1:500, DSHB, #9E10-C), mouse anti-beta tubulin (1:300, DSHB, #E7), rabbit anti-dsRed (1:200, Takara, #632496), Alexa488 goat anti-mouse (1:500, Invitrogen, #A11029), and Cy3 goat anti-rabbit (1:500, Jackson Immunoresearch, #111-165-003).

Fixed samples of Drosophila embryo nerve cords, mouse-dissociated commissural neurons, and mouse dorsal spinal cord explants were imaged using a spinning disk confocal system (Perkin Elmer) built on a Nikon Ti-U inverted microscope using a Nikon ×40 objective (for nerve cords and neurons)

and a ×10 objective (for explants) with Volocity imaging software. Images were processed using NIH ImageJ software.

## Electroporation of mouse embryos and primary neuron culture

E12.5 embryos were electroporated ex utero by injecting 100 ng/µl DNA in electroporation buffer (30 mM HEPES pH 7.5 [Thermo, #BP299-1], 300 mM KCl [Thermo, #BP366-1], 1 mM MgCl$_2$[Thermo, #BP214-500], and 0.1% Fast Green FCF [Thermo, #F99-10]) into the central canal of the neural tube. A BTX ECM 830 electroporator (BTX Harvard Apparatus, #45-0662) was used for bilateral electroporation into spinal cord neurons (five 30 V pulses, each of 50 ms duration for each half of the spinal cord). Following electroporation, dorsal spinal cords were dissected out and cut into explants for the explant outgrowth assay or used for preparation of dissociated neuronal cultures. For neuron culture, dissected spinal cords were washed in Hanks' Balanced Salt Solution (HBSS, Gibco, #14175-079) and digested with 0.05% trypsin (Gibco, #25300054) for 7 min at 37℃. 1 µl of DNase I (NEB, #M0303L) and 0.15% MgSO$_4$ (Thermo, #7487-88-9) was added for an additional minute, and the samples were centrifuged at 400 × g for 4 min. Samples were washed with pre-warmed HBSS, and a small fire-polished Pasteur pipette was used to triturate the tissue and dissociate it into single cells. Cells were plated on acid-washed, poly-D-lysine and N-cadherin-coated coverslips and cultured in plating media (Neurobasal [Gibco, #21103-049] medium supplemented with 10% heat-inactivated FBS and 1X penicillin/streptomycin/glutamine).

## Explant outgrowth assay

Dorsal spinal cord explants form E12.5 mouse embryos were dissected and cultured in collagen gels as previously described (Serafini 1994). Briefly, explants were embedded in rat tail collagen (Corning, #354249) gels at a distance of one explant diameter away from a mock 293 T cell aggregate (ATCC, CRL-3216) or a cell aggregate expressing Slit (pSecTagB-hSlit2-MYC, kind gift from A. Chedotal). Explants were grown in 50% OptiMEM and 45% Ham's F-12 media supplemented with 5% HS, 0.75% glucose, and 1X penicillin/streptomycin/glutamine for 48 hr with 500 ng/ml Netrin-1. Explants were subsequently fixed and stained as described above. For preparation of 293 T cell aggregates, cells were trypsinized and resuspended in a rat tail collagen solution, drawn into a glass Pasteur pipette, and allowed to polymerize. The collagen-embedded cells were released from the pipette using a rubber bulb and the aggregates cut into 1 mm clusters.

## Collapse assay

Dissociated commissural neurons from E12.5 mouse embryos were cultured in plating media (Neurobasal medium supplemented with 10% heat-inactivated FBS and 1X penicillin/streptomycin/glutamine) for 1 day in vitro. Plating media was replaced with Neurobasal supplemented with 1X B-27 for 1 hr. Neurons were treated with recombinant hSlit2-N (R&D, #5444-SL-050) at 2 µg/ml for 30 min at 37℃. Neurons were fixed immediately and immunostained for Robo3 (a marker for commissural neurons) and MYC to identify commissural neurons that had been successfully electroporated with the hRobo1-MYC or hRobo1ΔWIRS-MYC expression constructs.

## Chicken in ovo electroporation

Chicken in ovo electroporations were carried out as previously described (*Croteau et al., 2019*) at HH stage 14 and embryos were harvested at HH stage 22–23. Chicken embryos were electroporated with either pCAGGS, pCAGGS-hRobo1 wild-type or pCAGGS-hRobo1 WIRS-deletion constructs combined with pN2-eGFP at a 4:1 DNA weight ratio. Spinal cords were sectioned and stained with DAPI, anti-GFP (A11122, 1:5,000, Thermo Fisher) and anti-MYC (9E10, 1:200, DSHB) antibodies.

## Quantification and statistical analysis

For analysis of Drosophila nerve cord phenotypes, image analysis was conducted blind to the genotype. Data are presented as mean values ± SEM. For statistical analysis, comparisons were made between two groups using the Student's $t$-test. For multiple comparisons, significance was assessed using one-way ANOVA with Tukey's *post-hoc* tests. Differences were considered significant when $p < 0.05$. For quantitation of Scar intensity or surface HA intensity in Drosophila embryos, mean gray value for Scar or HA was obtained using ImageJ and normalized to the mean gray value for HRP.

Data are presented as mean values ± SEM. For statistical analysis, comparisons were made between two groups using the Student's *t*-test. Differences were considered significant when p<0.05. For the collapse assay, only Robo3-positive (and MYC-positive for neurons electroporated with hRobo1 variants) axons were imaged and analyzed. Growth cones were defined by the presence of lamellipodia and/or filopodia. Three trials were conducted and at least 30 neurons per condition were scored in each trial. Data are presented as mean values ± SEM. For statistical analysis, comparisons were made between groups using one-way ANOVA with Tukey's *post-hoc* tests. Differences were considered significant when p<0.05. To measure MYC signal intensity for total hRobo1 or surface hRobo1 quantitation in dissociated neurons, Robo3-positive neurons were carefully traced in ImageJ and integrated signal density in the traced region was obtained. Background signals were subtracted and mean fluorescence intensity calculated as integrated signal density per area is presented in graphs. Data are presented as mean values ± SEM. For statistical analysis, comparisons were made between two groups using the Student's *t*-test. Differences were considered significant when p<0.05. For the explant outgrowth assay, explants images were converted to black-and-white composites using the Threshold function. Each experimental set was quantified using the same threshold parameters. Explant quadrants were delineated by placing a right-angled crosshair at the center of each explant with the proximal quadrant directly facing the cell aggregate. The total area of black pixels was measured for the proximal and distal quadrants using the Analyze Particles function. The particles showing axonal outgrowth were then erased using the Eraser tool, and the total area of black particles was measured again. The difference was recorded as total area of axonal outgrowth. Next, the length of each quadrant was measured by tracing the border of the quadrant using the Freehand Line tool. Values for total area of outgrowth were normalized to length of the quadrant, and these final values were used to obtain the proximal/distal ratios for each explant. The measurements for each explant in a set were averaged, and the ratios of experimental conditions compared with control condition were calculated. Data are presented as mean ± SEM. Total number of explants for RFP control, RFP Slit, hRobo1 control, hRobo1 Slit, hRobo1ΔWIRS control, and hRobo1ΔWIRS Slit is 29, 39, 33, 39, 29, and 41, respectively (from three independent experiments). For statistical analysis, comparisons were made between groups using one-way ANOVA with Tukey's *post-hoc* tests. Differences were considered significant when p<0.05. For western blots, densitometric analysis was performed and band intensities of co-immunoprecipitating proteins in the immunoprecipitates were normalized to band intensities of HSPC300 in the immunoprecipitates as well as to lysate levels of the co-immunoprecipitating proteins. For each independent experiment, values were compared with wild-type Robo1 normalized values. Data are presented as mean ± SEM. For statistical analysis, comparisons were made between two groups using the Student's *t*-test. For multiple comparisons, significance was assessed using one-way ANOVA with Tukey's *post-hoc* tests. Differences were considered significant when p<0.05. For analysis of crossing index in chicken embryos, fluorescence intensities were generated by pixels above threshold using ImageJ. Five sections of each embryo were analyzed. For GFP crossing index, statistical significance was calculated using one-way ANOVA with post-hoc multiple comparisons. And for MYC crossing index, unpaired *t*-test was used. For all graphs, *p<0.05, **p<0.01, ***p<0.001, ****p<0.0001.

## Acknowledgements

We would like to acknowledge the members of the Bashaw Lab for discussions and comments on the manuscript. We thank Zachary DeLoughery and Alexander Jaworski for guidance on the in vitro explant experiments and Timothy Evans for the gift of the genomic Robo rescue construct. NSF Grant IOS-1853719 and NIH Grant R35 NS097340 to GJB supported this research.

## Additional information

### Funding

| Funder | Grant reference number | Author |
|---|---|---|
| National Institutes of Health | R35 NS097340 | Greg J Bashaw |
| National Science Foundation | IOS-1853719 | Greg J Bashaw |

The funders had no role in study design, data collection and interpretation, or the decision to submit the work for publication.

## Author contributions
Karina Chaudhari, Conceptualization, Data curation, Formal analysis, Investigation, Visualization, Methodology, Writing - original draft, Writing - review and editing; Madhavi Gorla, Data curation, Formal analysis, Investigation, Methodology; Chao Chang, Investigation, Methodology; Artur Kania, Supervision, Funding acquisition, Investigation, Methodology; Greg J Bashaw, Conceptualization, Resources, Supervision, Funding acquisition, Project administration, Writing - review and editing

## Author ORCIDs
Karina Chaudhari [iD] http://orcid.org/0000-0003-3533-3027
Artur Kania [iD] http://orcid.org/0000-0002-5209-2520
Greg J Bashaw [iD] https://orcid.org/0000-0002-6146-0962

## Ethics
Animal experimentation: This study was performed in strict accordance with the recommendations in the Guide for the Care and Use of Laboratory Animals of the National Institutes of Health. All of the animals were handled according to approved institutional animal care and use committee (IACUC) protocols of the Perelman School of Medicine at the University of Pennsylvania (Protocol #806216).

## Decision letter and Author response
Decision letter https://doi.org/10.7554/eLife.64474.sa1
Author response https://doi.org/10.7554/eLife.64474.sa2

# Additional files
## Supplementary files
• Transparent reporting form

## Data availability
All data generated or analysed during this study are included in the manuscript and supporting files. Confocal stacks were collected using a spinning disk confocal system (Perkin Elmer) built on a Nikon Ti-U inverted microscope with Volocity imaging software. Images were processed using NIH ImageJ and Adobe Photoshop software. All statistics and graphs were generated using GraphPad Prism 9.

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
