## [Decision Letter]

**Acceptance summary:**

The manuscript shows that Slit binding to Robo1 enhances the recruitment of the Scar/Wave regulatory complex (WRC) to the WIRS motif present the in Robo1 cytoplasmic tail. This interaction contributes to proper guidance of commissural axons in the *Drosophila* nerve cord and is conserved in vertebrates. The study provides novel insights in the mechanism of Slit/Robo signal transduction, especially important for axon guidance at the midline.

**Decision letter after peer review:**

Thank you for submitting your article "Robo recruitment of the Wave Regulatory Complex plays an essential and conserved role in midline repulsion" for consideration by *eLife*. Your article has been reviewed by 3 peer reviewers, one of whom is a member of our Board of Reviewing Editors, and the evaluation has been overseen by Marianne Bronner as the Senior Editor. The reviewers have opted to remain anonymous.

The reviewers have discussed the reviews with one another and the Reviewing Editor has drafted this decision to help you prepare a revised submission.

Summary:

This study identifies the WAVE regulatory complex (WRC) as a mediator of Robo receptor signaling that guides the trajectory of commissural axons in the *Drosophila* nerve cord. The authors show that the WIRS motif on the Robo receptor is required for this interaction and that ligand binding enhanced WRC recruitment to Robo. Mutual enhancement of midline guidance defects in combinatorial loss of function of slit/robo and WRC components and midline guidance defects seen in Robo receptors lacking the WIRS motif using both *Drosophila* and in vitro assays with mouse embryos, suggest conserved functional importance of this interaction in repulsive axon guidance.

Essential revisions:

The study is interesting and timely, given that the mechanisms that transduce axon guidance signaling are still not fully elucidated. Furthermore, Slit-Robo signaling operates well beyond axon guidance making these results attractive to a wide community. Nevertheless, there are several issues that needs to be addressed likely involving additional experimentation.

1. The relative importance of this novel arm of Robo signaling to axon guidance, with respect to other known downstream effectors such as Enabled and Rac, is unclear. The severity of the GOF phenotypes using the transgenic UAS-Robo1ΔWIRS allele in wild-type, and the rescue phenotypes in a robo null background (Figure 4), suggest an interaction with the WRC through WIRS is only partially responsible for midline repulsion, -indeed other downstream effectors of Robo could compensate for its loss in a dose dependent manner. This should be elucidated and discussed. Related to this point, IP with Abi and Nap1 may be interesting and could provide information on possible cooperation/interaction between effectors. Related to the latter, why have the author chosen to IP with hspc300? Why not using sra or cyfip? This should be specified. The IPs show there is a "constitutive" interaction between hspc300 and robo in basal (untreated condition), do the authors know if S2R+ cells express Slit ligand, that could be responsible for this interaction via autocrine pathway? Do the authors see any changes of cell shape or cytoskeleton organization after expression of Robo mutant or Slit treatment?

2. Related to the point above a demonstration that recruitment of the WRC to Robo through the WIRS motif indeed activates the WRC remains answered. Given that Rac can also activate the WRC, it is important to distinguish the effect of direct interaction with Robo through WIRS from other possible indirect Robo->Rac-> effects. It has been previously established that WRC activity is required for midline repulsion of commissural axons (Schenck et al., 2003), as is the role of the WIRS motif in WRC recruitment by Robo (Chen et al., 2014). Therefore, this present would be strengthened by a mechanistic dissection of this novel interaction- specifically its effect on WRC function and consequent changes to the actin cytoskeleton during repulsive axon guidance at the midline.

3. A major weakness is the presence of confounding deletions in the CRISPR-generated genomic Robo1ΔWIRS allele, complicating this analysis. The severe defects, comparable to a robo null, observed with the Robo1ΔWIRS genomic allele (Figure 5), indicate that interaction with WRC is essential for Robo mediated midline repulsion. However, the intronic deletion (Figure S5A) coupled to the conspicuous changes in the pattern of expression of this allele (Figures S5 C and F, compared to B and D) are important concerns, despite that protein levels appear to be normal. While it is possible that the deletion of WIRS is alone responsible for this altered localization, the presence of the confounding deletion complicates this interpretation. A Robo1ΔWIRS genomic allele devoid of any confounding modifications would be the best approach to drawing decisive conclusions here. If generation of a new allele is not underway, the authors may have additional evidence to offer or alternatively they need to discuss this weakness thoroughly.

4. The experiments shown in Figure 6 strictly show that the WIRS motive of Robo1 is required for Slit mediated axon repulsion. In principle the experiment does not really show that WRC per se is involved. Although, it seems easy to draw this conclusion, the authors should be careful in doing this extrapolation. In discussing this point they should also mention if there is anything known about WRCs and axon guidance in vertebrates or whether there are available mutant lines of WRC members for analysis of midline crossing. Are Robo1 WT and mutant forms expressed at equivalent levels in the explants?

5. Figures1 and 2: The same cyfipΔ85.1 homozygotes showed appreciable midline guidance defects in Schenck et al. 2003 and 2004a. In these studies, this was seen with the zygotic null, indicating that maternal contribution of Cyfip was insufficient to rescue zygotic LOF. What is the reason why the homozygotes show no phenotype on their own in this present study? This is important since these mutants are being used to demonstrate synergistic enhancements when combined with the slit; robo1/+ heterozygotes (Figure 1), or the sos, and robo2 homozygotes (Figure 1). If the lack of a phenotype in cyfip85.1 homozygotes on their own is due to the strain background used in this study, it is important that the slit; robo1/+ heterozygotes are also maintained on the same genetic background. If not, the observed enhancement could be attributed to the genetic background of the slit; robo/+ parents given that cyfip85.1/ cyfip85.1 alone is already known to show substantial midline guidance defects in at least one genetic background.

6. Figure 3: There is an inconsistency between experiments involving WRC-Robo affinity in the absence of Slit. In Figure 3 (C, E) WT Robo has significantly higher affinity for hspc300 compared to Robo1ΔWIRS in the absence of Slit. But in Figure 3 (D, F), WT Robo and Robo1ΔWIRS shown no significant difference in their affinity for WRC in the absence of Slit. This issue must be addressed. Also which is the explanation for Slit-mediated enhancement of Robo1-WRC interaction? Does ligand binding expose the WIRS domain or makes it more accessible?

---

## [Author Response]

Essential revisions:The study is interesting and timely, given that the mechanisms that transduce axon guidance signaling are still not fully elucidated. Furthermore, Slit-Robo signaling operates well beyond axon guidance making these results attractive to a wide community. Nevertheless, there are several issues that needs to be addressed likely involving additional experimentation.1. The relative importance of this novel arm of Robo signaling to axon guidance, with respect to other known downstream effectors such as Enabled and Rac, is unclear. The severity of the GOF phenotypes using the transgenic UAS-Robo1ΔWIRS allele in wild-type, and the rescue phenotypes in a robo null background (Figure 4), suggest an interaction with the WRC through WIRS is only partially responsible for midline repulsion, -indeed other downstream effectors of Robo could compensate for its loss in a dose dependent manner. This should be elucidated and discussed.

We apologize for not being clearer in our model for how the WIRS-WRC interaction feeds into the existing Robo1 signaling pathway. The reviewers are correct in noting the partial effects of disrupting the WIRS motif on Robo1 signaling in Figure 4. However, it is worth noting that the experiments in Figure 4 involved an overexpression of receptors. As we move to more endogenous levels of Robo1 (Figure 5), the effects of disrupting the WIRS motif on Robo1 signaling become more pronounced. We propose that the WRC is recruited to the WIRS motif in response to slit and is activated by Rac1. Active WRC can then act cooperatively with Ena to promote Arp2/3 meditated actin assembly. Indeed, Ena/VASP proteins have been shown to enhance WRC stimulation of Arp2/3 in the presence of Rac1 (Chen et al., 2014c). While indirect interactions with the WRC via Rac1 or Ena might confer some compensatory signaling, we propose that direct WIRS interaction would allow for localized actin polymerization and serve to anchor the complex to the site of ligand exposure thus allowing for tighter spatiotemporal control over subsequent downstream cytoskeletal changes. While it would be very interesting to further differentiate direct and indirect Rac1 effects, this would be particularly challenging to address via genetic interaction studies and is be beyond the scope of what we can achieve with this system. We thank the reviewers for pointing this out and we have edited the text to provide a better representation of our model.

Related to this point, IP with Abi and Nap1 may be interesting and could provide information on possible cooperation/interaction between effectors. Related to the latter, why have the author chosen to IP with hspc300? Why not using sra or cyfip? This should be specified. The IPs show there is a "constitutive" interaction between hspc300 and robo in basal (untreated condition), do the authors know if S2R+ cells express Slit ligand, that could be responsible for this interaction via autocrine pathway? Do the authors see any changes of cell shape or cytoskeleton organization after expression of Robo mutant or Slit treatment?

We thank the reviewers for these questions. The rationale for proceeding with HSPC300 for the co-IPs was driven primarily by availability of the construct and the relatively small size of the protein, which facilitates consistent levels of expression and reduces trial to trial variability. We have now specified this in the text. Relating to the point about a basal level of interaction, S2R+ cells do express low levels of slit (Figure 3—figure supplement 1D). It is quite possible that this basal level of slit secretion is responsible for the “constitutive” interaction noted in the absence of any application of slit-conditioned media. We have been unable to identify any gross cytoskeletal changes in S2R+ cells by staining for actin filaments using phalloidin.

2. Related to the point above a demonstration that recruitment of the WRC to Robo through the WIRS motif indeed activates the WRC remains answered. Given that Rac can also activate the WRC, it is important to distinguish the effect of direct interaction with Robo through WIRS from other possible indirect Robo->Rac-> effects. It has been previously established that WRC activity is required for midline repulsion of commissural axons (Schenck et al., 2003), as is the role of the WIRS motif in WRC recruitment by Robo (Chen et al., 2014). Therefore, this present would be strengthened by a mechanistic dissection of this novel interaction- specifically its effect on WRC function and consequent changes to the actin cytoskeleton during repulsive axon guidance at the midline.

We agree with the reviewers that further mechanistic dissection would strengthen the paper and we thank them for their suggestion. As mentioned above, we apologize for not being clearer in the text regarding our model for how the WIRS-WRC interaction feeds into the existing pathway and we refer the reviewers to the first point for clarification on this. As the reviewers correctly noted, the question of whether the WRC is activated downstream of Robo1 remains unanswered. In an effort to address this question, we performed additional experiments with members of the Arp2/3 complex, which is the primary downstream target of the WRC and can serve as a potential readout for WRC activation. We tested for genetic interactions between the Slit-Robo1 pathway and *arpc2*, a member of the Arp2/3 complex. While we see no Fas-II crossing errors in *arpc2* mutants alone, in the *slit, robo* sensitized background, *arpc2* mutants show a strong enhancement of the ectopic crossing defects (Figure 6A-C). Further, we see that *arpc2* mutants genetically interact with *cyfip* mutants in the *slit, robo* sensitized background resulting in a significant enhancement of the ectopic FasII crossing defects (Figure 6—figure supplement 1A-1C). These results suggest a cooperative effect of the WRC and Arp2/3 in the Slit-Robo1 repulsive pathway. Next, we overexpressed Robo1 in eagle neurons which results in a gain of function phenotype where almost all EW neurons fail to cross the midline. However, overexpressing Robo1 in *arpc2* homozygous mutants, results in a small but significant suppression of this phenotype that is similar to the suppression seen in *cyfip* mutants (Figure 6—figure supplement 1D-1I). Together, these genetic data strongly suggest that the Arp2/3 complex functions in the Slit Robo1 pathway.

We can also detect a physical interaction between Robo1 and Arp3, another component of the arp2/3 complex, in *Drosophila* S2R+ cells (Figure 6D). In contrast, mutating the WIRS motif substantially decreases this binding indicating that the interaction between Robo1 and Arp3 is partly dependent on the WIRS motif (Figure 6D and 6E). Additionally, we detect an increase in the interaction between Robo1 and Arp3 in the presence of Slit suggesting that similar to the WRC, the Arp2/3 is also recruited to Robo1 in response to Slit stimulation (Figure 6F and 6G). We reason that the WRC is activated downstream of Robo1 and is in turn responsible for the recruitment of the Arp2/3 complex to Robo1 to facilitate cytoskeletal remodeling.

3. A major weakness is the presence of confounding deletions in the CRISPR-generated genomic Robo1ΔWIRS allele, complicating this analysis. The severe defects, comparable to a robo null, observed with the Robo1ΔWIRS genomic allele (Figure 5), indicate that interaction with WRC is essential for Robo mediated midline repulsion. However, the intronic deletion (Figure S5A) coupled to the conspicuous changes in the pattern of expression of this allele (Figures S5 C and F, compared to B and D) are important concerns, despite that protein levels appear to be normal. While it is possible that the deletion of WIRS is alone responsible for this altered localization, the presence of the confounding deletion complicates this interpretation. A Robo1ΔWIRS genomic allele devoid of any confounding modifications would be the best approach to drawing decisive conclusions here. If generation of a new allele is not underway, the authors may have additional evidence to offer or alternatively they need to discuss this weakness thoroughly.

This is an important point brought up by the reviewers. We agree that the confounding deletion of the intron in the CRISPR mutant is not ideal and we have revised the text to clearly highlight the presence of this unintentional deletion and how it might complicate our interpretation. We believe that our rescue data showing complete rescue of the ectopic crossing defects in the CRISPR embryos using a genomic robo1 rescue construct that lacks this intron sequence, strongly argues against the idea that this intron is important for Robo1 signaling and we hope that the reviewers will agree. In addition, we would like to point out that several previous studies have shown that endogenous Robo function in axon guidance is completely independent of this intron (as well as all other intronic sequences), since replacing the Robo locus with an intron-less cDNA coding sequence completely substitutes for Robo function (See Spitzwek and Dickson, Cell 2010 and subsequent publications using these Robo knock-in alleles). Regarding the changes in the Robo1 expression pattern in this CRISPR mutant, some alteration in expression is expected due to the gross nerve cord morphology changes resulting from impaired Robo1 signaling. However, while it is interesting that we see Robo1 being expressed on commissures in these CRISPR embryos (Figure S5 C and F – now Figure 5—figure supplement 2D and 2G), it is not necessarily surprising to us as previous studies have also noted Robo1 expression on commissures when Robo1 signaling is disrupted (Fan et al., 2003; Coleman et al., 2010). Nevertheless, we have included a detailed description of these points in our discussion.

4. The experiments shown in Figure 6 strictly show that the WIRS motive of Robo1 is required for Slit mediated axon repulsion. In principle the experiment does not really show that WRC per se is involved. Although, it seems easy to draw this conclusion, the authors should be careful in doing this extrapolation. In discussing this point they should also mention if there is anything known about WRCs and axon guidance in vertebrates or whether there are available mutant lines of WRC members for analysis of midline crossing. Are Robo1 WT and mutant forms expressed at equivalent levels in the explants?

We agree with the reviewers. The experiments with vertebrate Robo1 support the importance of the WIRS motif but do not directly implicate the WRC and we have clarified this in the text. While knockout mouse lines are available for WRC members, given that there are multiple isoforms for WRC members in vertebrates, and that the WRC is involved in regulating several cellular processes, analysis of these mutants would be complicated and beyond the scope of the present study. For comparing levels of the Robo1 variants in explants, we co-electroporated the constructs with RFP and observed comparable levels of RFP staining in the explants (Figure 7—figure supplement 1A). We observed poor penetration of the MYC antibody through the explants embedded in collagen. Thus, in addition to RFP staining in explants, we also prepared commissural neuron cultures from the electroporated spinal cords and observed comparable levels of MYC staining between the two hRobo1 variants in dissociated neurons (Figure 7—figure supplement 1B and 1C). Additionally, to ensure that both hRobo1 variants are being trafficked to the surface comparably, we assayed surface expression of the MYC-tagged receptors and found that both forms were comparably expressed on the surface of dorsal commissural neurons (Figure 6—figure supplement 2C and 2D.

5. Figures1 and 2: The same cyfipΔ85.1 homozygotes showed appreciable midline guidance defects in Schenck et al. 2003 and 2004a. In these studies, this was seen with the zygotic null, indicating that maternal contribution of Cyfip was insufficient to rescue zygotic LOF. What is the reason why the homozygotes show no phenotype on their own in this present study? This is important since these mutants are being used to demonstrate synergistic enhancements when combined with the slit; robo1/+ heterozygotes (Figure 1), or the sos, and robo2 homozygotes (Figure 1). If the lack of a phenotype in cyfip85.1 homozygotes on their own is due to the strain background used in this study, it is important that the slit; robo1/+ heterozygotes are also maintained on the same genetic background. If not, the observed enhancement could be attributed to the genetic background of the slit; robo/+ parents given that cyfip85.1/ cyfip85.1 alone is already known to show substantial midline guidance defects in at least one genetic background.

This is a good question. The reviewers are correct in noting that we do not see significant guidance defects in these *cyfip* mutants, contrary to those reported by Schenck *et al.* in 2003 and 2004a. Given the fact that it has been two decades since these studies were published, it is likely that genetic modifiers have accumulated to mask this phenotype. Nevertheless, the fact that cyfip mutants show a striking enhancement of the slit, robo phenotype and that this effect can be rescued cell-autonomously by expressing a cyfip transgene argues strongly against any contribution of strain differences to the phenotypes we observe. Indeed, the fact that these mutants in our hands have no phenotypes on their own argues even more strongly for the significance of the genetic interactions that we observe. Thus, considering the UAS-CYFIP rescue data in apterous neurons (Figure 1Q) and the fact that the *cyfip* allele was crossed into the *slit, robo* sensitized background, we are confident that the defects seen are not due the strain background.

6. Figure 3: There is an inconsistency between experiments involving WRC-Robo affinity in the absence of Slit. In Figure 3 (C, E) WT Robo has significantly higher affinity for hspc300 compared to Robo1ΔWIRS in the absence of Slit. But in Figure 3 (D, F), WT Robo and Robo1ΔWIRS shown no significant difference in their affinity for WRC in the absence of Slit. This issue must be addressed. Also which is the explanation for Slit-mediated enhancement of Robo1-WRC interaction? Does ligand binding expose the WIRS domain or makes it more accessible?

We thank the reviewers for pointing this out. For Figure 3 (C, E), the interaction experiments were performed in the absence of application of any conditioned media whereas in Figure 3 (D, F), the cells were treated with either mock conditioned media or slit-conditioned media. Thus, it is likely that addition of mock conditioned media results in low levels of receptor activation which might explain the different observed affinities in E and F. Indeed, we can detect low levels of slit in concentrated mock conditioned media (Figure 3—figure supplement 1D). We have clarified this point in the figure legends and clearly indicated the addition of mock conditioned media in the graph in F. As for the question of how slit facilitates the Robo1-WRC interaction, whether it results in exposure of the WIRS motif or simply increases accessibility to it, we cannot exclude either possibility from our current data.